# DENSITY-BASED CLUSTERING WITH KERNEL DIFFUSION

## ABSTRACT

Finding a suitable density function is essential for density-based clustering algorithms such as DBSCAN and DPC. A naive density corresponding to the indicator function of a unit $d$-dimensional Euclidean ball is commonly used in these algorithms. Such a density suffers from an inability to capture local features in complex datasets. To tackle this issue, we propose a new kernel diffusion density function, which is adaptive to data of varying local distributional characteristics and smoothness. Furthermore, we develop a surrogate that can be efficiently computed in linear time and space and prove that it is asymptotically equivalent to the kernel diffusion density function. Extensive empirical experiments on benchmark and large-scale face image datasets show that the proposed approach not only achieves a significant improvement over classic density-based clustering algorithms but also outperforms the state-of-the-art face clustering methods by a large margin.

## 1 INTRODUCTION

Density-based clustering algorithms are now widely used in a variety of applications, ranging from high energy physics (Tramacere & Vecchio, 2012; Rovere et al., 2020), material sciences (Marquis et al., 2019; Reza et al., 2007), social network analysis (Shi et al., 2014; Khatoon & Banu, 2019) to molecular biology (Cao et al., 2017; Ziegler et al., 2020). In these algorithms, data points are partitioned into clusters that are considered to be sufficiently or locally high-density areas with respect to an underlying probability density or a similar reference function. We call them density functions throughout this paper. These techniques are attractive to practitioners, due to their non-parametric feature, which leads to flexibility in discovering clusters that have arbitrary shapes, whilst classic methods such as $k$-means and $k$-medoids (Friedman et al., 2001) can only detect convex (e.g., spherical) clusters. Seminal work in the context of density-based clustering includes DBSCAN (Ester et al., 1996) and DPC (Rodriguez & Laio, 2014), among many others (Ankerst et al., 1999; Cuevas et al., 2001; Comaniciu & Meer, 2002; Hinneburg & Gabriel, 2007; Stuetzle, 2003).

Most density-based clustering algorithms implicitly identify cluster centers and assign remaining points to the clusters by connecting with the higher density point nearby. To proceed with these methods it requires a density function, which is usually an estimate of the underlying true probability density or some variants of it. For example, a popular choice is the naive density function that is carried out by simply calculating the number of data points covered in the $\varepsilon$-neighborhood of each $x$. Note that such densities are not adaptive to different distribution regions. One of the challenging scenarios is when clusters in the data have varying local features, for example, size, height, spread, and smoothness. Therefore, the resulting density function has a tendency to flatten the peaks and valleys in the data distribution, which leads to underestimation of the number of clusters (see Figure 1). Many heuristics variations of DBSCAN and DPC have been proposed to magnify the local features, thus making the clustering task easier (Campello et al., 2013; Chen et al., 2018; Ertöz et al., 2003; Zhu et al., 2016). Most of these methods can be viewed as performing clustering on certain transformations of the naive density function. However, if the naive density function itself is quite problematic in the first place, these methods will become less effective.

Moreover, even if we apply adaptive alternatives to modify the classic density functions, there are other contentious issues of the generally used linear kernel density estimator (KDE). It often suffers from severe boundary bias (Marron & Ruppert, 1994) and is acknowledged as computationally

Figure 1: (a) Data generated from Gaussian mixture model with 3 components, each has differing variance and weight. (b) Naive density function in 2D (top) and 3D (bottom): only one peak can be identified. (c) Proposed kernel diffusion density function: 3 clusters can be easily discovered.

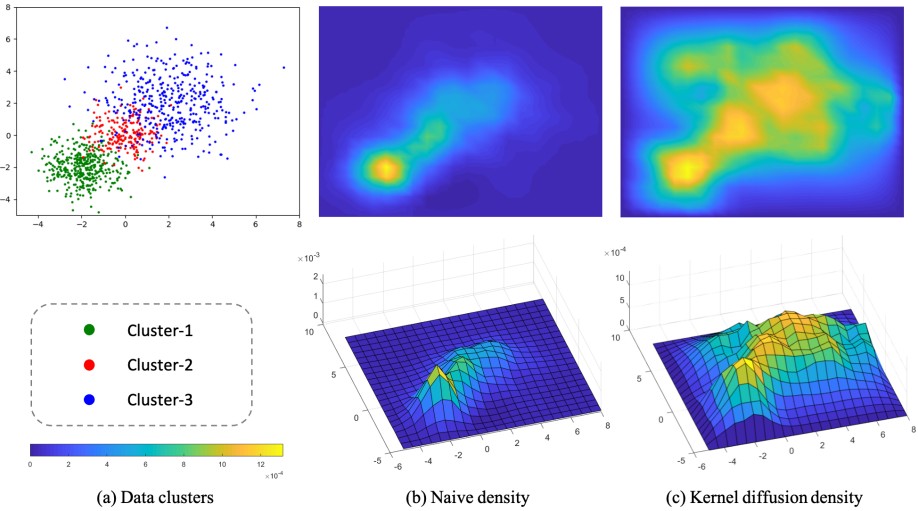

(a) Data clusters  (b) Naive density  (c) Kernel diffusion density

expensive. These phenomenon prevent the classic density functions being practically useful and reliable, especially for large-scale and complex clustering tasks.

To overcome these problems in density-based clustering algorithms, in this paper we propose a general approach to build the so-called kernel diffusion density function to replace classic density functions. The key idea is to construct the density from a user-specified bivariate kernel that has desired local adaptive properties. Instead of using the naive density function and its variants, we utilize the bivariate kernel to derive a transition probability. A diffusion process is induced by this transition probability, which admits a limiting and stationary distribution. This limiting distribution serves as a plausible density function for clustering with reduced error.

Under this framework, we provide examples of symmetric and asymmetric bivariate kernels to construct the kernel diffusion density function, which can tackle clustering complex and locally varying data. We apply the resulting adapted DBSCAN and DPC algorithms to widely different empirical datasets and show significant improvement in each of these analyses. The main contributions of this paper are summarized below:

- We introduce new bivariate kernel functions and construct the associated kernel diffusion processes. Based on the diffusion process, we propose a kernel diffusion density function to adapt density-based clustering algorithms such as DBSACN and DPC, which attains accuracy in the presence of varying local features.

- We derive a computationally much more efficient surrogate, and show analytically it is asymptotic equivalent to the proposed kernel diffusion density function.

- By extensive experiments, we demonstrate the superiority of kernel diffusion density function over naive density function and its variants when applying to DBSCAN and DPC, and show it outperforms state-of-the-art GCN-based methods on face clustering tasks.

## 2 RELATED WORK

**Density-Based Clustering**    There is vast literature on adapting density-based clustering algorithms to tackle large variations in different clusters in the data. DPC itself is such a refinement of DBSCAN, as it determines cluster centers not only by highest density values but also by taking into account their distances from each other, thus has a generally better performance in complex clustering tasks. Other attempts include rescaling the data to have relative reference measures instead of KDE (Zhu

et al., 2016; Chen et al., 2018), and using the number of shared-nearest-neighbors between two points to replace the geometric distance (Ertöz et al., 2003).

**Diffusion Maps** The technique of diffusion maps (Coifman et al., 2005; Coifman & Lafon, 2006) gives a multi-scale organization of the data according to their underlying geometric structure. It uses a local similarity measure to create a diffusion process on the data which integrates local geometry at different scales along time $t$. Generally speaking, the diffusion will segment the data into several smaller clusters in small $t$ and group data into one cluster for large $t$. Applying eigenfunctions at a carefully selected time $t$ leads to good macroscopic representations of the data, which is useful in dimension reduction and spectral clustering (Nadler et al., 2005).

**Face Clustering** Face clustering has been extensively studied as an important application in machine learning. Traditional algorithms include $k$-means, hierarchical clustering (Sibson, 1973) and ARO (Otto et al., 2017). Many recent works take advantage of supervised information and GCN models, achieving impressive improvement comparing to traditional algorithms. To name a few, CDP (Zhan et al., 2018) proposes to aggregate the features extracted by different models; L-GCN (Wang et al., 2019) predicts the linkage in an instance pivot subgraph; LTC (Yang et al., 2019) generates a series of subgraphs as proposals and detects face clusters thereon; and GCN(V+E) (Yang et al., 2020) learns both the confidence and connectivity by GCN. In this paper we demonstrate that the proposed density-based clustering algorithm with kernel diffusion, as a general clustering approach, even outperforms theses state-of-the-art methods that are especially designed for face clustering.

## 3 PRELIMINARIES

### 3.1 NOTATIONS

Let the dataset $D = \{x_1, \ldots, x_n\} \subset \mathbb{R}^d$ be $n$ i.i.d samples drawn from a distribution measure $F$ with density $f$ on $\mathbb{R}^d$. Let $F_n$ denote the corresponding empirical distribution measured with respect to $D$, i.e., $F_n(A) = \frac{1}{n} \sum_{i=1}^{n} \mathbf{1}_A(x_i)$, where $\mathbf{1}_A(\cdot)$ denotes the indicator function of set $A$. We write $||u||$ as the Euclidean norm of vector $u$. Let $B(x, \varepsilon)$ and $V_d$ denote the $d$-dimensional $\varepsilon$-ball centered at $x$ and the volume of the unit ball $B(0, 1)$, respectively. Let $N_k(x)$ denote the set of $k$-nearest neighbors of point $x$ within the dataset $D$.

### 3.2 DENSITY FUNCTION

Density-based algorithms perform clustering by specifying and segmenting high-value areas in a density function denoted by $\rho$. Usually, we calculate each of $\rho(x_i)$, and then identify cluster centers with (locally) highest values. Many popular algorithms such as DBSCAN and DPC employ the following naive density function:

$$\rho_{\text{naive}}(x) = \frac{1}{n\varepsilon^d} \sum_{y \in D} \frac{\mathbf{1}_{B(x,\varepsilon)}(y)}{V_d}. \tag{1}$$

The naive density function $\rho_{\text{naive}}$ is actually an empirical estimation of $f$ for carefully chosen $\varepsilon$. It is easy to observe, for clustering purpose we only care about $\rho_{\text{naive}}(x)$ up to a normalising constant, which makes it simply equivalent to counting the total number of data points in the $\varepsilon$-ball around $x$.

In practice, the data distribution may be very complex and contains varying local features that are difficult to be detected. The naive density in (1) with the same radius $\varepsilon$ for all $x$ usually suffers from unsatisfactory empirical performance, for example, failing to identify small clusters with fewer data points. One possible way to alleviate this problem is through a transformation into the following local contrast (LC) function (Chen et al., 2018):

$$\rho_{\text{LC}}(x) = \frac{1}{n} \sum_{y \in N_k(x)} \mathbf{1}_{\rho_{\text{naive}}(x) > \rho_{\text{naive}}(y)}. \tag{2}$$

In this way, $\rho_{\text{LC}}$ compares the density of each data point with its $k$-nearest neighbors. To see the benefit of LC, let us consider $x$ to be a cluster center. After local contrasting, $\rho_{\text{LC}}(x)$ is likely to reach the value of $k$ regardless of the size of this cluster.

However density functions like $\rho_{\text{LC}}$ still highly depend on the underpinning performance of $\rho_{\text{naive}}$. This restricts their applications in clustering data with challenging local features.

## 4 METHODOLOGY

In this section, we present a new type of density-based clustering algorithm, based on the notion of kernel diffusion density function. Towards this end, we will introduce a kernel diffusion density function, which takes account of local adaptability and is well-suited for clustering purpose. We provide details on how to derive this density function from a diffusion process induced by bivariate kernels. We also provide a surrogate density function that is computationally more efficient.

### 4.1 DIFFUSION PROCESS AND KERNEL DIFFUSION DENSITY

Considering a bivariate kernel function $k : D \times D \to \mathbb{R}^+$, such that:

- $k(x, y)$ is positive semi-definite, i.e., $k(x, y) \geq 0$.
- $k(x, y)$ is $F_n$-integrable with respect to both $x$ and $y$.

We define $d(x) = n \int_D k(x, y) dF_n(y)$ as a local measure of the volume at $x$ and define

$$p(x, y) = \frac{k(x, y)}{d(x)}. \tag{3}$$

It is easy to see that $p(x, y)$ satisfies the conservation property $n \int_D p(x, y) dF_n(y) = 1$. As a result, $p(x, y)$ can be viewed as a probability for a random walk on the dataset from point $x$ to point $y$, which induces a Markov chain on $D$ with $n \times n$ transition matrix $P = [p(x, y)]$. This technique is standard in various applications, known as the normalized graph Laplacian construction. For example, we can view $D$ as a graph, $L = I - P$ as the normalized graph Laplacian, and $d(x)$ as a normalization factor.

For $t \geq 0$, the probability of transiting from $x$ to $y$ in $t$ time steps is given by $P^t$, the $t$-th power of $P$. Running the Markov chain forward in time, we observe the dataset at different scales, which is the diffusion process $X_t$ on $D$. Let $\rho(x, t): D \times \mathbb{R}^+ \to \mathbb{R}^+$ be the associated probability density, which is governed by the following second-order differential equation with initial conditions:

$$\begin{cases} \frac{\partial}{\partial t} \rho(x, t) = -L\rho(x, t), \\ \rho(x, 0) = \phi_0(x), \end{cases} \tag{4}$$

where $\phi_0(x)$ is a probability density at time $t = 0$. In practice we can use any valid choice of $\phi_0(x)$, e.g. the uniform density.

To give an explicit example of the diffusion process, consider a sub-class of $k$, i.e., isotropic kernels, where $k(x, y) = \mathcal{K}(||x - y||^2 / h)$ for some function $\mathcal{K} : \mathbb{R} \to \mathbb{R}^+$. Here we can dual interpret $h$ as a scale parameter to infer local information and as a time step $h = \Delta t$ at which the random walk jumps. Then we can define the forward Chapman-Kolmogorov operator $T_F$ as

$$T_F(x) = n \int_D p(x, y) \phi_0(y) dF_n(y).$$

Note that $T_F$ is the data distribution at time $t = h$, thus can be viewed as continuous analogues of the left multiplication by the transition matrix $P$. Letting $h \to 0$, the random walk converges to a continuous diffusion process with probability density evolves continuously in $t$. In this case, we can explicitly write the second-order differential equation in (4) as:

$$\frac{\partial}{\partial t} \rho(x, t) = \lim_{h \to 0} \frac{\hat{\rho}(x, t + h) - \rho(x, t)}{h} = \lim_{h \to 0} \frac{T_F - I}{h} \rho(x, t), \tag{5}$$

where $L_h = \lim_{h \to 0} (T_F - I)/h$ is the conventional infinitesimal generator of the process.

Now we are ready to introduce our kernel diffusion density function.

**Definition 1.** *(Kernel diffusion density function) Suppose the Markov chain induced by $P$ is ergodic, we define the kernel diffusion density function as the limiting probability density of the diffusion process $X_t$, i.e.,*

$$\rho_{\mathrm{KD}}(x) = \lim_{t \to \infty} \rho(x, t). \tag{6}$$

Intuitively, on the one hand, with increased values of $t$ we expect the diffusion process $X_t$ gradually reveals the geometric structure (such as high-density regions) of the data distribution $F$. To see this, note that the transition probability $P$ reflects connectivity between data. We can interpret a cluster as an underlying geometric structure in which the probability of staying in this region is high during a transition. In the diffusion process, the probability of following a path along a structure usually increases with $t$, as the involved data points are dense and highly connected. Therefore, the path consists of short and high probability jumps. Whilst paths that do not follow any structure consists of long and low probability jumps, which lowers their overall probability. As a result, geometry structures of $F$ is magnified during the diffusion.

On the other hand, by talking certain sophisticated forms of $k(x, y)$ that take into account of local adaptivity, we also slow down the diffusion to avoid trivial geometry structures such as one big cluster for all the data points. In this way, we can eventually identify the correct geometry structure at the right scale.

## 4.2 Locally Adaptive Kernels

To address the local adaptability in kernel diffusion density function, we propose the following two bivariate kernels. Both of them are very simple variations of the most commonly used classic kernels.

**Symmetric-Gaussian kernel:**

$$k(x, y) = \exp\left( -\frac{\|x - y\|^2}{h} \right) \mathbf{1}_{B(x, \varepsilon)}(y). \tag{7}$$

Here $h$ and $\epsilon$ are both hyper-parameters. We call this kernel symmetric since $k(x, y) = k(y, x)$ .

**Asymmetric-Gaussian kernel:**

$$k(x, y) = \exp\left( -\frac{\|x - y\|^2}{h} \right) \mathbf{1}_{N_k(x)}(y). \tag{8}$$

Here $h$ and $k$ are hyper-parameters. Note that in this case $k(x, y)$ is asymmetric as $y \in N_k(x)$ does not imply $x \in N_k(y)$.

Bivariate kernels defined in (7) and (8) are just combinations of classic Gaussian kernel and $\varepsilon$-neighbourhood or $k$-nearest neighbours kernels, respectively. With these simple combinations, we truncate Gaussian kernel at local areas, and the contribution of each point $y$ to the construction of the density function $\rho_{\mathrm{KD}}(x)$ depends not only on the distance $\|y - x\|$ but also on the local geometry structure around $x$. Hence, the new kernels are adaptive at different $x$, which is expected to lead to better clustering performance against local features. We remark that the Asymmetric-Gaussian kernel takes into account a varying neighborhood around each $x$, thus is more adaptive comparing to the Symmetric-Gaussian kernel.

Although here we only provide two examples of locally adaptive kernels, other options can be easily created in a similar spirit under this framework, e.g., changing the Gaussian kernels to other kernels or changing the $\varepsilon$-neighbourhood ($k$-nearest neighbours) kernels to other locally truncated functions. Once $k(x, y)$ is determined, we can derive the corresponding density function $\rho_{\mathrm{KD}}$. Next, we just need to simply apply any density clustering procedure like DPC or DBSCAN based on $\rho_{\mathrm{KD}}$ instead of the naive density function $\rho_{\mathrm{naive}}$.

In Section 5, we assess the empirical performance of the proposed kernel diffusion density function with the above two locally adaptive kernels. They outperform existing density-based algorithms and other state-of-the-art methods.

### 4.3 FAST KERNEL DIFFUSION DENSITY

The kernel diffusion density function $\rho_{\text{KD}}$ can be calculated as the stationary distribution of a Markov chain induced by the transition matrix $P$. Numerically, we can solve it by iteratively right multiplying $P$ with $\rho(x, t)$ until convergence, or applying a QR decomposition on $P$. These methods are expensive in terms of computational cost, especially when the sample size $n$ is large.

To tackle this problem, we propose the following surrogate of $\rho_{\text{KD}}(x)$ which is computationally more efficient.

**Definition 2.** *(Fast kernel diffusion density function) Let $p(y, x)$ be the transition probability from point $y$ to point $x$, as defined in equation (3). We define the fast kernel diffusion density function as*

$$\rho_{\text{FKD}}(x) = \int_D p(y, x) dF_n(y), \tag{9}$$

It is straightforward that $\rho_{\text{FKD}}$ can be obtained in linear time and memory space, as we only need to compute the column averages of matrix $P$.

Here we show that $\rho_{\text{FKD}}$ is not only computationally efficient but also suitable for detecting local features. This is illustrated through the following Theorem 1. Consider a special case that $k(x, y) = \mathbf{1}_{B(x,\varepsilon)}(y)$. Then it is easy to verify that

$$\rho_{\text{FKD}}(x) = \frac{1}{C_d} \sum_{y \in B(x,\varepsilon)} \frac{1}{\rho_{\text{naive}}(y)},$$

where $C_d = n\varepsilon^d V_d$ is a normalising constant. In this way, we build a connection between $\rho_{\text{FKD}}$ and the naive density function $\rho_{\text{naive}}$ in this special example.

**Theorem 1.** *Consider the above special case that $k(x, y) = \mathbf{1}_{B(x,\varepsilon)}(y)$. In addition, assume the dataset $D = \{x_1, ..., x_n\}$ can be split into $m$ disjoint clusters: i.e., $X = D_1 \bigcup ... \bigcup D_m$ and for each $x \in D$, $B(x, \varepsilon)$ only contain data points that belong to the same cluster as $x$. Denote $\bar{\rho}_j = \frac{1}{|D_j|} \sum_{x \in D_j} \rho_{\text{FKD}}(x)$ as the average density in cluster $j$. We have*

$$\bar{\rho}_1 = \cdots = \bar{\rho}_m = 1.$$

Theorem 1 demonstrates that the averaged $\rho_{\text{FKD}}$ in each cluster are the same regardless of cluster sizes and other local features. This shows that $\rho_{\text{FKD}}$ elevates the density of small clusters, which is essential for finding the density peaks of small clusters.

Previously we claim that $\rho_{\text{FKD}}$ is a surrogate of the kernel diffusion density $\rho_{\text{KD}}$. Next, we want to reveal the relationship between these two density functions from an asymptotic viewpoint. To proceed, we will need the following assumption.

**Assumption 1.** *There exists some positive constant $c < 1$ that is independent of $n$, such that $\rho_{FKD}(x) \leq c$ holds for every $x \in D$.*

This is a very mild assumption, since it always holds that $\rho_{\text{FKD}}(x) < 1$, and the average of $\rho_{\text{FKD}}(x)$ over the dataset is $\int_D \rho_{\text{FKD}}(x) dF_n(x) = 1/n$, which vanishes as $n \to \infty$. Now we are ready to present the following theorem that characterise the closeness between $\rho_{\text{FKD}}$ and $\rho_{\text{KD}}$.

**Theorem 2.** *Suppose that Assumption 1 holds and the Markov chain induced by the kernel $k(x, y)$ is ergodic. We have*

$$\frac{\rho_{KD}(x)}{\rho_{FKD}(x)} \xrightarrow{a.s.} 1$$

As shown in the Appendix, the almost sure convergence in Theorem 2 is of a fast rate at $n^{-1}$. Thus it is safe for us to use it to replace $\rho_{\text{KD}}$ in finite sample experiments. This result is also verified by our numerical experiments in Section 5.

## 5 EXPERIMENTS

In this section, we empirically evaluate the proposed kernel diffusion density functions against $\rho_{\text{naive}}$ and $\rho_{\text{LC}}$ in density-based clustering algorithms, and also compare them with other state-of-the-art

Table 1: Clustering performance on benchmark datasets with different density functions applied to DPC. Pairwise F-score ($F_P$) and BCube F-score ($F_B$) under optimal parameter tuning are given. The best and second-bset results in each dataset are bolded and underlined, respectively.

| Dataset | $F_P$ | | | | | | $F_B$ | | | | | |
|---|---|---|---|---|---|---|---|---|---|---|---|---|
| | $\rho_{\text{naive}}$ | $\rho_{\text{LC}}$ | $\rho_{\text{KD}}^{\text{sym}}$ | $\rho_{\text{KD}}^{\text{asym}}$ | $\rho_{\text{FKD}}^{\text{sym}}$ | $\rho_{\text{FKD}}^{\text{asym}}$ | $\rho_{\text{naive}}$ | $\rho_{\text{LC}}$ | $\rho_{\text{KD}}^{\text{sym}}$ | $\rho_{\text{KD}}^{\text{asym}}$ | $\rho_{\text{FKD}}^{\text{sym}}$ | $\rho_{\text{FKD}}^{\text{asym}}$ |
| Banknote | 54.3 | 31.6 | 67.2 | 83.9 | 67.2 | **93.6** | 57.7 | 31.8 | 67.2 | 85.1 | 67.2 | **93.6** |
| Breast-d | 55.9 | 51.8 | **78.0** | 69.1 | 67.4 | 72.6 | 59.0 | 58.7 | **76.0** | 69.7 | 69.4 | 72.2 |
| Breast-o | 57.6 | 70.7 | 82.8 | **92.9** | 82.7 | **92.9** | 52.2 | 74.1 | 75.9 | 92.2 | 75.8 | **92.2** |
| Control | 48.6 | 49.3 | 49.0 | 63.9 | 52.5 | **64.5** | 51.6 | 52.4 | 52.0 | 70.8 | 55.1 | **71.8** |
| Glass | 36.9 | 39.0 | 46.3 | **48.1** | 44.8 | 47.8 | 42.7 | 45.7 | 55.1 | 56.9 | 53.5 | **57.1** |
| Haberman | 66.9 | 64.1 | 74.5 | 75.7 | **75.8** | 75.7 | 66.9 | 63.3 | 74.5 | 75.8 | 75.9 | 75.8 |
| Ionosphere | 27.4 | 28.3 | 46.9 | **54.9** | 46.0 | 53.9 | 25.0 | 25.8 | 42.6 | **52.5** | 41.7 | 49.2 |
| Iris | 54.3 | 53.8 | 65.8 | **74.6** | 69.2 | **74.6** | 61.6 | 62.3 | 72.7 | **80.0** | 74.0 | **80.0** |
| Libras | 20.0 | 22.9 | 29.3 | **31.5** | 26.0 | 31.0 | 26.8 | 29.1 | 37.8 | **41.8** | 33.3 | 39.8 |
| Pageblocks | 92.9 | **93.0** | 90.5 | 90.2 | 89.7 | 90.2 | 89.9 | **90.0** | 89.8 | 89.7 | 89.6 | 89.7 |
| Seeds | 54.3 | 54.9 | 68.0 | **78.0** | 69.5 | **78.0** | 54.3 | 55.4 | 72.4 | **78.7** | 72.9 | **78.7** |
| Segment | 48.4 | 48.0 | 57.1 | **58.0** | 41.4 | 56.1 | 64.2 | 63.8 | 67.1 | **69.2** | 60.6 | 68.2 |
| Wine | 45.2 | 61.1 | 56.6 | **68.0** | 60.0 | 65.3 | 46.0 | 61.9 | 61.5 | **74.7** | 66.3 | 71.4 |

methods. We denote $\rho_{\text{KD}}^{\text{sym}}$ and $\rho_{\text{FKD}}^{\text{sym}}$ as the kernel diffusion density functions and its fast surrogate, with symmetric-Gaussian kernel, respectively. Similarly, we denote $\rho_{\text{KD}}^{\text{asym}}$ and $\rho_{\text{FKD}}^{\text{asym}}$ as the proposed two density functions with asymmetric-Gaussian kernel, respectively. We examine their performance on a wide range of datasets. The clustering results are measured Pairwise F-score (Banerjee et al., 2005), BCubed F-score (Amigó et al., 2009) or NMI (Cover, 1999).

## 5.1 PERFORMANCE ON BENCHMARK DATASETS

We now discuss the performance on 13 benchmark datasets ($\sim$100 to $\sim$5,000 data points) from UCI repository. The metadata is summarised in the Appendix.

As summarised in Table 1, both $\rho_{\text{KD}}^{\text{sym}}$ and $\rho_{\text{KD}}^{\text{asym}}$ uniformly outperform $\rho_{\text{naive}}$ and $\rho_{\text{LC}}$ in terms of clustering accuracy in terms of F-scores. results based on NMI are deferred to the Appendix. The proposed kernel diffusion density functions with asymmetric Gaussian kernel, $\rho_{\text{KD}}^{\text{asym}}$, which enjoys better local adaptivity analytically, achieves the best results on most datasets and outperforms $\rho_{\text{naive}}$ and $\rho_{\text{LC}}$ by a large margin. It is worth noticing that the two fast surrogates, $\rho_{\text{FKD}}^{\text{sym}}$ and $\rho_{\text{FKD}}^{\text{asym}}$, achieve comparable results with their original counterparts, $\rho_{\text{KD}}^{\text{sym}}$ and $\rho_{\text{KD}}^{\text{asym}}$. In the Appendix similar results are observed for the same set of density functions applied to DBSCAN.

Figure 2: Precision-Recall curves of different approaches applied to DPC on MS1M dateset, using (a) Pairwise metric, and (b) BCubed metric .

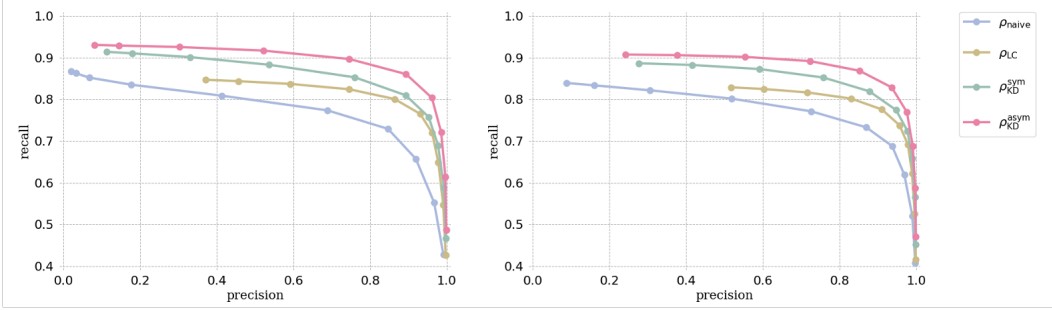

## 5.2 PERFORMANCE ON FACE IMAGE DATASETS

Clustering face images according to their latent identity becomes an important application in recent years. It is challenging in the sense that face image datasets usually contain thousands of identities, corresponding to thousands of clusters. Meanwhile, the number of images for each identity (cluster) is quite different, corresponding to the variety of cluster sizes. We assess the performance of the proposed approach on two popular face image datasets: emore_200k (Zhan et al., 2018) and MS1M (Guo et al., 2016).

**emore_200k.** The dataset contains 2,577 identities with 200,000 images following the protocol in Zhan et al. (2018). We set $\varepsilon$ to 0.8, $k$ to 200, and $h$ to 0.5 for density-based methods. Results are summarized in Table 2. The proposed diffusion density functions are applied to DPC, and compared with $k$-means, HAC (Sibson, 1973), ARO (Otto et al., 2017), and CDP (Zhan et al., 2018). Again, we observe significant improvement in the proposed density functions over $\rho_{\text{naive}}$ and $\rho_{\text{LC}}$. It is also worth pointing out that, density-based clustering with proposed kernel diffusion density functions also outperform the state-of-the-arts approaches such as CDP by a large margin.

Table 2: Clustering performance on emore_200k. BCubed precision, recall and F-score are reported.

| | Algorithm | # clusters | Precision | Recall | $F_B$ |
|---|---|---|---|---|---|
| Baseline | $k$-means | 2,577 | 94.24 | 74.89 | 83.45 |
| | HAC | 2,577 | **97.74** | 88.02 | 92.62 |
| | ARO | 85,150 | 52.96 | 16.93 | 25.66 |
| | CDP | - | 89.35 | 88.98 | 89.16 |
| Density -based | $\rho_{\text{naive}}$ | 7928 | 92.36 | 78.14 | 84.65 |
| | $\rho_{\text{LC}}$ | 3485 | 96.15 | 86.58 | 91.11 |
| | $\rho_{\text{KD}}^{\text{sym}}$ | 2781 | 95.82 | 93.24 | 94.51 |
| | $\rho_{\text{KD}}^{\text{asym}}$ | 2546 | 95.48 | 93.82 | 94.64 |
| | $\rho_{\text{FKD}}^{\text{sym}}$ | 3622 | 95.27 | 92.54 | 93.89 |
| | $\rho_{\text{FKD}}^{\text{asym}}$ | 2569 | 96.37 | **93.93** | **95.13** |

- Not available

**MS1M.** The dataset contains 8,573 identities with around 584,000 images following the protocols in Yang et al. (2020). We set $\varepsilon$ to 0.8, $k$ to 200, and $h$ to 0.5 for density-based methods. We reported the results of clustering performance in Table 3. Precision versus Recall curves for different density functions (applied to DPC) are plotted in Figure 2. In Table 3, the proposed kernel diffusion density functions outperform $\rho_{\text{naive}}$ and $\rho_{\text{LC}}$. Note that GCN-based methods such as L-GCN (Wang et al., 2019), LTC (Yang et al., 2019) and GCN (V+E) (Yang et al., 2020) achieve generally better clustering performance than unsupervised methods due to their supervised nature. However, it is quite encouraging to see that the proposed kernel diffusion approaches, although are also unsupervised clustering methods, considerably outperform the GCN-based methods.

Table 3: Clustering performance on MS1M. Pairwise F-score and BCubed F-score are reported.

| | Algorithm | # clusters | $F_P$ | $F_B$ |
|---|---|---|---|---|
| Unsupervised | $k$-means | 8,573 | 79.21 | 81.23 |
| | HAC | 8,573 | 70.63 | 70.46 |
| | ARO | - | 13.60 | 17.00 |
| | CDP | - | 75.02 | 78.70 |
| Supervised | L-GCN | - | 78.68 | 84.37 |
| | LTC | - | 85.66 | 85.52 |
| | GCN(V+E) | - | 87.55 | 85.94 |
| Density-based | $\rho_{\text{naive}}$ | 59551 | 78.37 | 79.35 |
| | $\rho_{\text{LC}}$ | 24019 | 83.61 | 85.06 |
| | $\rho_{\text{KD}}^{\text{sym}}$ | - | - | - |
| | $\rho_{\text{KD}}^{\text{asym}}$ | 22869 | **88.15** | 87.14 |
| | $\rho_{\text{FKD}}^{\text{sym}}$ | 34246 | 84.40 | 85.37 |
| | $\rho_{\text{FKD}}^{\text{asym}}$ | 22927 | 87.26 | **87.41** |

- Not available

## 5.3 SENSITIVITY ANALYSIS

Next, we examine the sensitivity of the proposed kernel diffusion density functions to hyper-parameters and compare it with $\rho_{\text{naive}}$ and $\rho_{\text{LC}}$. The results are obtained via extensive experiments on emore_200k and MS1M, which are shown in Figure 3. We can see that the clustering performance of $\rho_{\text{KD}}^{\text{sym}}$ is much more stable than $\rho_{\text{naive}}$ and $\rho_{\text{LC}}$ when we vary the value of $\varepsilon$. Whilst $\rho_{\text{KD}}^{\text{asym}}$ is robust to the parameter $k$, and both $\rho_{\text{KD}}^{\text{sym}}$ and $\rho_{\text{KD}}^{\text{asym}}$ are quite robust to the parameter $h$.

Figure 3: Sensitivity analysis on emore_200k and MS1M. We investigate the clustering performance by varying the following parameters: (a) Radius of $\varepsilon$-ball; (b) Number $k$ of nearest neighbors; (c) Bandwidth $h$ of Gaussian kernel.

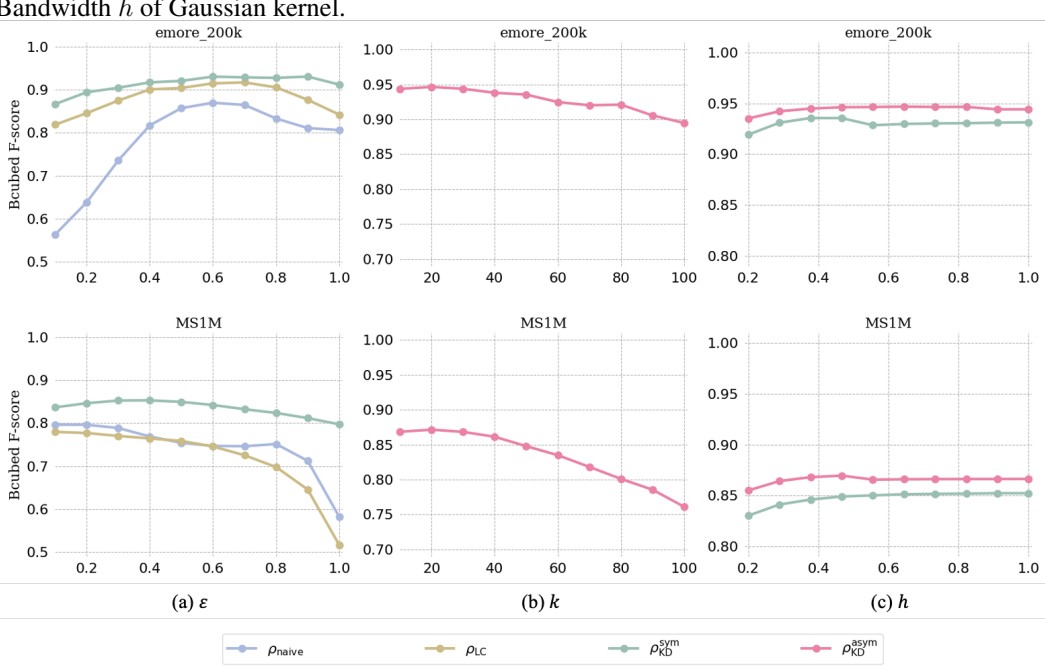

## 5.4 COMPUTATIONAL COST

We carried out a series of experiments on MS1M to demonstrate the computational efficiency of the fast surrogate $\rho_{FKD}$ in terms of time and space. With a collection of subsampled data from MS1M at different percentile levels, we run both the kernel diffusion density $\rho_{KD}$ and the fast surrogate $\rho_{FKD}$. As we can observe from Figure 4, the running time and memory usage of $\rho_{KD}$ increase dramatically with the sample size. Whilst $\rho_{FKD}$ retains a very low level of computational cost. This suggests that $\rho_{FKD}$, which achieves an excellent computational efficiency, should be favored in practice.

Figure 4: Running time and memory usage of the proposed methods at different sample sizes on MS1M.

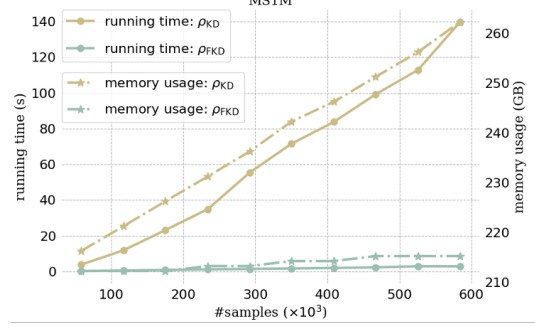

## 6 CONCLUSION

Density-based clustering has a profound impact on machine learning and data mining. However, the underpinning naive density function suffers from detecting varying local features, causing extra errors in the clustering. We propose a new set of density functions based on the kernel diffusion process to resolve this problem, which is adaptive to density regions of varying local distributional features. We demonstrate that DBSCAN and DPC adapted by the proposed approach have improved clustering performance comparing to their classic versions and other state-of-the-art methods.

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

## A APPENDIX

In this supplementary file, we provide technical proofs of the theoretical results in Section 4.3, and present extra empirical experiments regarding our kernel diffusion approach with symmetric and asymmetric Gaussian kernels applied to DBSCAN. All the numerical experiments are carried out on a standard work station with a Intel 64-cores CPU and two Nvidia P100 GPUs.

### A.1 PSEUDO CODE FOR DBSCAN AND DPC.

---
**Algorithm 1** DBSCAN
---
1: **Input:** SetOfPoints $X$, Eps $\varepsilon$, MinPts $k$
2: $H := \{x \in X : |B(x, \varepsilon) \cap X| \geq k\}$;
3: $G :=$ undirected graph with vertices $H$ and edge between $x, x' \in H$ if $|x - x'| \leq \varepsilon$;
4: **Output:** connected components of $G$

---

The connected compoenents of the graph are determined a clusters,a dn the remaining points are unclustered and considered as noise-points.

---
**Algorithm 2** DPC
---
1: **Input:** SetOfPoints, TruncDis $d_c$
2: Compute $d_{i,j}$ for $\forall i, j \in$ SetOfPoints;
3: For $i$ from 1 to SetOfPoints.size:
4: $\quad\quad \rho_i := \sum_{i \neq j} \mathbb{1}_{\{d_{i,j} - d_c\}}$
5: $\quad\quad \delta_i := min_{j:\rho_j > \rho_i}(d_{i,j})$
6: Plot decision map $M$ with $\rho$ as the horizontal axis and $\delta$ as the vertical axis;
7: Mark Point $i$ with relatively higher $\rho_i$ and $\delta_i$ as a cluster center;
8: Mark Point $i$ with relatively lower $\rho_i$ but relatively higher $\delta_i$ as a noise point;
9: Assign the rest point with the label the same as the nearest cluster center;
10: **Output:** SetOfPoints

---

### A.2 PROOFS OF THEORETICAL RESULT.

**Proof of Theorem 1.** Since $\{D_1, \cdots, D_m\}$ are disjoint, we have $p(x, y) = 0$ if $x$ and $y$ belong to different clusters. By the definition of matrix $P$, for each $x \in D_j$, we have

$$\int_D p(x, y) dF_n(y) = 1,$$

which implies that

$$\int_{x \in D_j} \int_{y \in D_j} p(x, y) dF_n(x) dF_n(y) = |D_j|.$$

Therefore,

$$\bar{\rho}_j |D_j| = \int_{x \in D_j} \int_{y \in D_j} p(x, y) dF_n(x) dF_n(y) = |D_j|,$$

which implies that $\bar{\rho}_j = 1$ for any $j = 1, , \ldots, m$. $\qquad\square$

Before proceeding to the proof of Theorem 2, we need following auxiliary lemma that relates the stationary distribution of a Markov chain to an arbitrary vector $g$.

**Lemma A.1.** *Let $P$ be transition probability matrix of a finite inreducible discrete time Markov chain with $n$ states, which admits a stationary distribution, denoted by vector $\pi$. We write $e = (1, \ldots, 1)^T \in \mathbb{R}^n$ as the a column vector of ones. The following holds for any vector $g$ such that $g^T e \neq 0$:*

*(1) $(I - P + eg^T)$ is non-singular.*

*(2) Let $H = (I - P + eg^T)^{-1}$, then $\pi^T = g^T H$.*

*Proof.* Since $\pi$ is the stationary distribution, we have $\pi^T e = 1$. Applying Theorem 3.3 in (Hunter, 1982) yields that matrix $(I - P + eg^T)$ is non-singular.

Next recall that $\pi^T P = \pi^T$, therefore we have

$$\begin{aligned}
\pi^T(I - P + eg^T) &= \pi^T - \pi^T P + \pi^T eg^T \\
&= \pi^T eg^T \\
&= g^T,
\end{aligned}$$

which implies $\pi^T = g^T H$. □

**Proof of Theorem 2.** Note that for for each $x \in D$, the linear reference function $\rho_{\text{FKD}}(x) = \int_D p(y, x) dF_n(y)$ is the corresponding column average of the transition matrix $P$. We write the $i$-th column vector of $P$ as

$$p_i = \big(p(x_1, x_i), \ldots, p(x_n, x_i)\big)^T.$$

Therefore $\rho_{\text{FKD}}(x_i) = e^T p_i / n$

Since the Markov chain induced by the kernel $k(x, y)$ is ergodic, the density $\rho(x, t)$ of the diffusion process $X_t$ converges to the limiting stationary distribution of the Markov chain, denoted by $\pi$.

We can write the $n$-vectors of $g$ and $\pi$ in the following form:

$$g = (g_1, \ldots, g_n)^T = n\big(\rho_{\text{FKD}}(x_1), \ldots, \rho_{\text{FKD}}(x_n)\big)^T \quad \text{and} \quad \pi = \big(\rho_{\text{KD}}(x_1), \ldots, \rho_{\text{KD}}(x_n)\big)^T,$$

where $g_i = e^T p_i$ is the $i$-th column sums of matrix $P$. As a result, we have

$$\int_D \rho_{\text{FKD}}(x) dF_n(x) = \frac{1}{n} e^T g = 1, \quad \text{and} \quad n \int_D \rho_{\text{KD}} dF_n(x) = e^T \pi = 1.$$

By the definition of $g$, we know

$$(eg^T)^2 = neg^T \quad \text{and} \quad e^T P = g^T.$$

It follows from Lemma A.1 that $(I - P + eg^T)$ is non-singular and $\pi^T = g^T H$, where $H = (I - P + eg^T)^{-1}$.

We define $B = I + eg^T$. By simple algebra calculation, we can find $B$ is non-singular with

$$B^{-1} = I - \frac{eg^T}{n+1}.$$

As a result, it is easy to see that that $g^T B^{-1} = \frac{g^T}{n+1}$ and

$$H^{-1} = B - P = (I - PB^{-1})B.$$

Use the Neumann series, we have

$$H = B^{-1}(I - PB^{-1})^{-1} = B^{-1} \sum_{i=0}^{\infty} (PB^{-1})^i.$$

Thus

$$\pi^T - g^T/n = g^T\left(H - \frac{I}{n}\right) = g^T\left[B^{-1}\sum_{i=0}^{\infty}(PB^{-1})^i - \frac{I}{n}\right].$$

Since we assume for any $x \in D$, $\hat{g}(x) < c$ for some $0 < c < 1$. This leads to

$$g^T p_j \leq nce^T p_j = ncg_j.$$

Therefore, let $\kappa_j$ be the $j$-th compoeent of $g^T PB^{-1}$, it is straightforward

$$\kappa_j \leq \frac{nc}{n+1} g_j \leq cg_j.$$

This implies for every $x \in D$,

$$|\rho_{\text{KD}}(x) - \rho_{\text{FKD}}(x)| \leq \rho_{\text{FKD}}(x) \left| \frac{1}{n+1} \sum_{i=0}^{\infty} c^i - \frac{1}{n} \right|$$

$$\leq \rho_{\text{FKD}}(x) \left| \frac{1}{(n+1)(1-c)} - \frac{1}{n} \right|.$$

Hence we have

$$\mathbb{P}\left( \lim_{n \to \infty} \frac{\rho_{\text{KD}}(x)}{\rho_{\text{FKD}}(x)} = 1 \right) = 1,$$

which completes the proof. □

### A.3 ADDITIONAL NUMERICAL EXPERIMENTS

**Naive density with different bandwidths.**     To illustrate the fail of naive density function in scenario as in Figure 1, we also plot it with a range of different values of hyperparameters $\epsilon$ below. We can observe that it is difficult to detect the three true underlying clusters in all the cases.

Figure 5: Naive density function in 3D with different values of $\epsilon$.

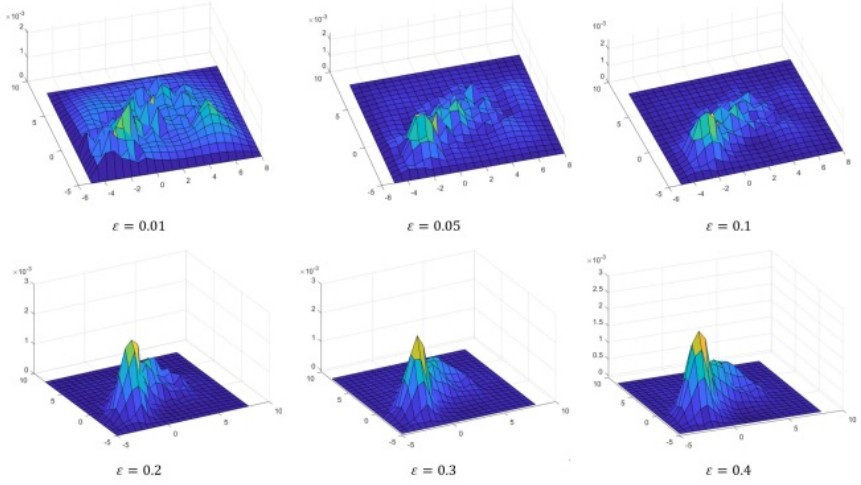

**Hyperparameters**     The parameter $\varepsilon$ (radius of the ball, used in $\rho_{\text{naive}}$, $\rho_{\text{LC}}$, $\rho_{\text{KD}}^{\text{sym}}$ and $\rho_{\text{FKD}}^{\text{sym}}$) is tuned by searching within the range between 0.1 and 1 with am increment of 0.1, parameter $k$ (number of nearest neighbors, used in $\rho_{\text{LC}}$, $\rho_{\text{KD}}^{\text{asym}}$ and $\rho_{\text{FKD}}^{\text{asym}}$) is tuned by searching within the range between 10% and 50% number of samples, with an increment of 10%.

**Metadata of benchmark datasets.**     The number of samples $n$, the number of clusters $c$, and feature dimension $d$ for each benchmark dataset are listed in Table 4 below.

**Benckmark datasets with DBSCAN.**     We provide the performance of the conventional density functions, $\rho_{\text{naive}}$ and $\rho_{\text{LC}}$, and the proposed kernel diffusion density functions with symmetric and asymmetric Gaussian kernels, $\rho_{\text{KD}}^{*}$ and $\rho_{\text{FKD}}^{*}$ ($* \in \{\text{sym}, \text{asym}\}$), applied to DBSCAN on 13 benchmark datasets. The results are summarised in Table 6. Similar to DPC, we see that both $\rho_{\text{KD}}^{\text{sym}}$ and $\rho_{\text{KD}}^{\text{asym}}$ uniformly outperform $\rho_{\text{naive}}$ and $\rho_{\text{LC}}$ in terms of clustering quality. $\rho_{\text{KD}}^{\text{asym}}$, which has better local adaptivity analytically, achieves the best results on most datasets and outperforms others by a significant margin in Breast-o, Control, Haberma and Seeds.

**NMI for benchmark datasets.**     Below we present in Table 6 the clustering results for benchmark datasets based on NMI metric.

Table 4: Metadata of benchmark datasets, includes sample size ($n$), the number of clusters ($c$), and feature dimension $d$.

| Dataset | $n$ | $c$ | $d$ |
|---|---|---|---|
| Banknote | 1372 | 2 | 4 |
| Breast-d | 569 | 2 | 30 |
| Breast-o | 699 | 2 | 9 |
| Control | 600 | 6 | 60 |
| Glass | 214 | 7 | 9 |
| Haberman | 306 | 2 | 3 |
| Ionosphere | 351 | 2 | 34 |
| Iris | 150 | 3 | 4 |
| Libras | 360 | 15 | 90 |
| Pageblocks | 5473 | 5 | 10 |
| Seeds | 210 | 3 | 7 |
| Segment | 210 | 7 | 19 |
| Wine | 178 | 3 | 13 |

Table 5: Clustering performance on benchmark datasets with different density functions applied to DBSCAN. Pairwise F-score ($F_P$) and BCube F-score ($F_B$) under optimal parameter tuning are given. The best and second-best results in each dataset are bolded and underlined, respectively.

| Dataset | $F_P$ | | | | | | $F_B$ | | | | | |
|---|---|---|---|---|---|---|---|---|---|---|---|---|
| | $\rho_{\text{naive}}$ | $\rho_{\text{LC}}$ | $\rho_{\text{KD}}^{\text{sym}}$ | $\rho_{\text{KD}}^{\text{asym}}$ | $\rho_{\text{FKD}}^{\text{sym}}$ | $\rho_{\text{FKD}}^{\text{asym}}$ | $\rho_{\text{naive}}$ | $\rho_{\text{LC}}$ | $\rho_{\text{KD}}^{\text{sym}}$ | $\rho_{\text{KD}}^{\text{asym}}$ | $\rho_{\text{FKD}}^{\text{sym}}$ | $\rho_{\text{FKD}}^{\text{asym}}$ |
| Banknote | 26.8 | 60.7 | 62.0 | **66.4** | 65.4 | **66.4** | 26.5 | 65.1 | 60.7 | **67.4** | 65.7 | **67.4** |
| Breast-d | 56.7 | 63.0 | 65.0 | 66.6 | **67.2** | 66.6 | 60.9 | 64.7 | 66.0 | 67.4 | **67.2** | 67.4 |
| Breast-o | 18.2 | 55.3 | 59.2 | **70.6** | 70.5 | 70.6 | 15.9 | 50.7 | 52.3 | **71.3** | 70.6 | 71.2 |
| Control | 32.5 | 37.1 | 51.0 | **60.3** | 48.9 | 59.1 | 34.2 | 50.1 | 53.7 | **66.9** | 51.6 | 65.7 |
| Glass | 22.0 | 29.8 | 29.8 | **42.5** | 42.0 | **42.5** | 25.8 | 36.9 | 36.9 | **45.2** | 43.5 | **45.2** |
| Haberman | 68.6 | 72.2 | 68.6 | **75.6** | 68.9 | 75.3 | 69.2 | 73.1 | 69.2 | **75.8** | 68.3 | 75.7 |
| Ionosphere | 25.9 | 68.4 | 68.0 | **74.2** | **74.2** | **74.2** | 23.8 | 64.1 | 63.7 | **72.1** | **72.1** | **72.1** |
| Iris | 66.2 | 69.8 | 66.2 | 57.2 | **73.7** | 73.3 | 67.2 | 76.6 | 67.2 | 67.0 | **79.4** | 79.0 |
| Libras | 18.1 | 12.0 | 15.6 | 13.8 | **20.2** | 13.5 | 31.1 | 16.5 | 42.1 | **45.5** | 32.9 | 37.7 |
| Pageblocks | 48.4 | 89.2 | 90.0 | **90.1** | 89.9 | **90.1** | 45.2 | 85.5 | **89.7** | 89.5 | **89.7** | 89.5 |
| Seeds | 57.8 | 47.6 | 57.8 | **63.2** | 22.4 | 62.4 | 59.2 | 53.0 | 59.2 | **70.0** | 24.4 | 69.2 |
| Segment | 18.5 | 47.9 | **54.8** | 30.8 | 41.4 | 30.8 | 22.5 | 55.2 | **66.6** | 53.6 | 59.9 | 53.6 |
| Wine | 40.5 | 40.5 | 40.5 | 49.5 | **50.0** | 49.5 | 45.7 | 45.7 | 45.7 | **52.3** | 51.3 | **52.3** |

**Number of clusters.** In Table 7, we present the number of clusters returned by the density-based methods for the benchmark datasets. It can be observed that clustering with the proposed diffusion density functions returned a significantly better estimate of the number of clusters, comparing to that with classic density functions such as $\rho_{naive}$ and $\rho_{LC}$.

Table 6: Clustering performance on benchmark datasets with different density functions applied to DPC. NMI under optimal parameter tuning are given. The best results in each dataset are bolded.

| Dataset | NMI | | | | | | | |
| --- | --- | --- | --- | --- | --- | --- | --- | --- |
| | $\rho_{\text{naive}}$ | $\rho_{\text{LC}}$ | $\rho_{\text{KD}}^{\text{sym}}$ | $\rho_{\text{KD}}^{\text{asym}}$ | $\rho_{\text{FKD}}^{\text{sym}}$ | $\rho_{\text{FKD}}^{\text{asym}}$ | $k$-means | Spectral |
| Banknote | 27.5 | 33.0 | 21.7 | 64.8 | 53.2 | **80.2** | 34.2 | 17.3 |
| Breast-d | 43.7 | 49.1 | 46.8 | 57.4 | 55.7 | 46.1 | **62.3** | 52.6 |
| Breast-o | 30.2 | 32.7 | 37.2 | **79.1** | 36.4 | 78.4 | 74.8 | 14.0 |
| Control | 60.6 | 60.6 | 63.2 | 69.5 | 61.0 | 69.6 | **75.4** | 68.3 |
| Glass | 43.1 | 43.4 | 45.0 | **48.4** | 43.8 | 46.6 | 34.8 | 36.4 |
| Haberman | 9.5 | 5.7 | 9.5 | 3.2 | **16.9** | 3.2 | 7.8 | 6.6 |
| Ionosphere | 27.9 | 28.0 | 30.9 | **31.1** | 30.1 | 30.5 | 13.5 | 5.2 |
| Iris | 51.1 | 53.1 | 60.1 | 73.4 | 62.6 | 73.4 | **74.2** | 70.6 |
| Libras | 63.3 | 66.4 | 63.0 | 68.8 | 68.2 | **69.1** | 60.0 | 56.1 |
| Pageblocks | 8.6 | 13.0 | 11.8 | 28.7 | 14.4 | **29.1** | 13.2 | 12.1 |
| Seeds | 47.1 | 49.8 | 53.6 | 64.8 | 58.6 | 64.8 | **67.4** | 60.3 |
| Segment | 63.5 | 64.4 | 65.1 | **72.2** | 63.0 | 70.7 | 61.2 | 65.2 |
| Wine | 58.1 | 58.2 | 72.0 | 73.3 | 71.1 | 58.6 | **84.2** | 72.7 |

Table 7: Number of clusters returned by different density functions applied to DPC. The ground truth is listed in the last column.

| Dataset | $\rho_{\text{naive}}$ | $\rho_{\text{LC}}$ | $\rho_{\text{KD}}^{\text{sym}}$ | $\rho_{\text{KD}}^{\text{asym}}$ | $\rho_{\text{FKD}}^{\text{sym}}$ | $\rho_{\text{FKD}}^{\text{asym}}$ | Ground Truth |
| --- | --- | --- | --- | --- | --- | --- | --- |
| Banknote | 16 | 44 | 26 | 2 | 7 | 2 | 2 |
| Breast-d | 3 | 5 | 2 | 3 | 3 | 3 | 2 |
| Breast-o | 15 | 17 | 7 | 2 | 1 | 2 | 2 |
| Control | 32 | 32 | 23 | 23 | 27 | 25 | 6 |
| Glass | 4 | 21 | 4 | 8 | 3 | 7 | 7 |
| Haberman | 34 | 40 | 34 | 1 | 3 | 1 | 2 |
| Ionosphere | 12 | 11 | 7 | 6 | 7 | 7 | 2 |
| Iris | 8 | 12 | 4 | 2 | 3 | 2 | **3** |
| Libras | 12 | 60 | 2 | 61 | 100 | 55 | 15 |
| Pageblocks | 7 | 25 | 3 | 10 | 11 | 8 | 5 |
| Seeds | 10 | 10 | 2 | 6 | 3 | 6 | **3** |
| Segment | 21 | 27 | 14 | 8 | 6 | 9 | 7 |
| Wine | 3 | 7 | 3 | 5 | 3 | 2 | **3** |

