# OpenReview forum: "Density-based Clustering with Kernel Diffusion"
_ICLR.cc/2022/Conference — ICLR 2022 Submitted_

### Official Review · Reviewer_SSTf · 2021-10-27

**Correctness:** 3
**Technical Novelty And Significance:** 3
**Empirical Novelty And Significance:** 2
**Recommendation:** 5
**Confidence:** 5

**Main Review:**

The proposed idea is interesting and novel. However, there are several concerns to be addressed:
-There is no theoretical justification that the proposed method is able to capture the local characteristics of the dataset.
-The fast surrogate (FKD) density is very simple to compute. It seems strange that it provides comparable performance to KD. Are there cases where FKD does not perform well?
-What is the complexity of computing \rho_{KD}?
- A major drawback of the method is that it includes additional hyperparameters compared to the naïve approach. In the presented experimental results (Table 1) the value of hyperparameter h=0.5. Therefore the empirical conclusions are conditioned on this value.
- Clustering is an unsupervised problem. It is not acceptable to tune the hyperparameters based on supervised measures.
- I suggest that the authors should provide NMI values for the datasets of Table 1. This is the typical measure. For the face datasets that are more complex, the use of F-measures is acceptable. Moreover, for the datasets of Table 1, it is suggested to present results using typical clustering methods (eg. k-means) as happens with the face datasets.
-  It is important to provide information on the number of clusters returned by the density-based methods for all datasets. Do they recover the ground truth number of clusters?
- What is the dimensionality of the face datasets?
- As a general remark, density-based methods are not considered effective in the case of small datasets of relatively high dimensionality.


**Summary Of The Paper:**

The paper presents a methodology for defining the density reference function required in density-based clustering methods such as DBSCAN and DPC. The density function is obtained as the limiting probability density of a diffusion process and it takes into account the local characteristics of the dataset. A surrogate density is also proposed that is fast to compute. The method is compared to typical density functions used in DBSCAN and DPC.

**Summary Of The Review:**

The proposed idea is interesting and suggests an alternative way for specifying density functions for density-based clustering. However, there exist several unresolved issues regarding theoretical justification and, mainly, experimental validation.

---

> ### Author Response · Authors · 2021-11-16
> **Response to Reviewer SSTf**
>
> We understand the reviewer's concerns, and hope the following answers provide a satisfying response.
>
> **There is no theoretical justification that the proposed method is able to capture the local characteristics of the dataset**
>
> Although we are very keen to establish rigorous theoretical results to support the proposed method in terms of clustering, this is not a quite feasible task. Note that the theoretical consistency theory for DBSCAN is developed in [1] and follow-up works, on a simplified version of the naive-DBSCAN. For naive-DPC, even it has been widely used in enormous applications, analysis of its theoretical properties is still not available to the best of our knowledge.  Fairly speaking, we believe this requirement is far beyond the scope of this paper intended for ICLR.
>
>
> The purpose of Theorem 1 is actually to provide a simple justification. For the naïve approach, density is calculated as the number of data points covered in the $\varepsilon$-nerghborhood. Since the $\varepsilon$ is the same for all data points, the density of points in large and compact clusters will be much larger than that of points in small and spread clusters, which makes it difficult to discover these small and loose clusters. In the special scenario introduced in Theorem 1, no matter the varying distribution of different clusters, the average densities of all clusters are always comparable using our kernel diffusion approach. This lead to better clustering results as we can now easily identify small clusters.
>
> **Are there cases where $\rho_{\text{FKD}}$ does not perform well? What is the complexity of computing $\rho_{\text{KD}}$?**
>
> As shown in Theorem 2, $\rho_{\text{FKD}}$ and $\rho_{\text{KD}}$ attain the same accuracy when the sample size $n$ is sufficient large. In large datasets such as face image clustering data,  $\rho_{\text{FKD}}$ has comparable performance as $\rho_{\text{KD}}$ . The difference of $\rho_{\text{FKD}}$ and $\rho_{\text{FKD}}$ becomes more significant in small datasets. The complexity of computing $\rho_{\text{KD}}$ is $O(n^2)$.
>
>
> **The method includes additional hyperparameters compared to the naïve approach. In the presented experimental results (Table 1) the value of hyperparameter $h=0.5$. Therefore the empirical conclusions are conditioned on this value.**
>
> The additional hyperparameter is not a particular problem, as our methods do not rely on estimating the density via KDE, where bandwidth selection is very sensitive and crucial. For example, see the sensitivity analysis of $h$ in Figure 3, the clustering results are very stable for a wide range of $h$.
>
> **Clustering is an unsupervised problem. It is not acceptable to tune the hyperparameters based on supervised measures**
>
> Just like k-means and spectral clustering that requires the number of clusters as input, there are tuning parameters in density-based clustering methods such as DBSCAN and DPC. Tuning these hyperparameters properly is a necessary step to evaluate the clustering performance of all these methods.
>
> In $\rho_{naïve}$, $\rho_{LC}$, $\rho_{KD}$ and $\rho_{FKD}$, all hyperparameters were tuned in the same manner, just as in the paper of local contrast density [2], which is one of our main competitors. Furthermore, in face image clustering, deep learning-based methods such as CDP, L-GCN, LTC, and GCN(V+E) are all tuned based on supervised measures. This is a common practice in existing works and we carried out experiments in the same way for a fair comparison.
>
> Additionally,  sensitivity analysis is presented in Figure 3, where We can see the hyperparameters used in $\rho_{KD}$ and $\rho_{FKD}$ are less sensitive compared to other methods. And in this way, the selection of hyperparameters in our method is much easier than other density-based methods.
>
> **I suggest that the authors should provide NMI values for the datasets of Table 1. This is the typical measure. For the more complex face datasets, the use of F-measures is acceptable. Moreover, for the datasets of Table 1, it is suggested to present results using typical clustering methods (eg. k-means) as happens with the face datasets**
>
> Thanks for your suggestion. We now provide NMI values in the appendix.
>
>
> **What is the dimensionality of the face datasets? As a general remark, density-based methods are not considered effective in the case of small datasets of relatively high dimensionality**
>
> The dimensionality of face datasets is 256. Since the sample sizes of face image datasets are usually large, density-based clustering methods are suitable.
>
>
> [1] *Consistency and rates for clustering with dbscan." Artificial Intelligence and Statistics*. Sriperumbudur, B., & Steinwart, I, PMLR, 2012.
>
>
> [2] *Local contrast as an effective means to robust clustering against varying densities.Machine Learning*.  Chen, B., etc.  Machine Learning, 2018

---

> > ### Author Response · Authors · 2021-11-19
> > **We look forward to your further feedback!**
> >
> > Dear Reviewer SSTf,
> >
> > Thanks again for your constructive suggestions and comments. As the deadline of discussion is approaching, we would like to know if there is any additional clarification or explanation that you may need, and we will be happy to provide them accordingly.
> >
> > In our previous response, we have studied your comments and addressed them carefully. We summarized the main points below:
> > 1. We provided  explanation on the possibility of theoretical justifications for density-based clustering.
> > 2. We conducted the experiments on benchmark datasets, using NMI as you suggested
> > 3. We provided additional explanation on the hyperparameter tuning issues. You may also want to have a look at our discussoins with Reviewer Nbci regarding similar issues.
> >
> > Please do not hesitate to let us know if there is additional response we can offer.
> >
> > Thank you for your time and effort spent in reviewing our paper!

---

> > > ### Comment · Reviewer_SSTf · 2021-11-20
> > > **Response to response**
> > >
> > > Thank you for the response and the revision.
> > >
> > > As far as I can see, you have not responded to the  following question:
> > > 'It is important to provide information on the number of clusters returned by the density-based methods for all datasets. Do they recover the ground truth number of clusters?'
> > >
> > > As it can be observed in Table 6, k-means provides better NMI results in 5 out 13 datasets which is a significant percentage.

---

> > > > ### Author Response · Authors · 2021-11-20
> > > > **Thank you very much for your prompt follow up comments!**
> > > >
> > > > Please see our response below. We also revised the paper to include additional information on number of clusters as you suggested.
> > > >
> > > > **As far as I can see, you have not responded to the following question: 'It is important to provide information on the number of clusters returned by the density-based methods for all datasets. Do they recover the ground truth number of clusters?'**
> > > >
> > > > Thank you for your very helpful suggestion. In the revised paper (see Tables 2 and 3), we now provide the number of clusters returned by the density-based methods for the face image datasets.  The proposed methods returned significantly better estimates of the number of clusters,  comparing to classic density-based methods such as $\rho_{naive}$ and $\rho_{LC}$. The estimates are quite close the ground truth. As we are carrying out experiments based on  optimal parameter tuning, methods such as $k$-means and HAC, which directly include the number of clusters as a tuning parameter, will not superisingly recover the ground truth number of clusters exactly. However, their clusetering performance are not as good as the proposed methods.
> > > >
> > > >
> > > > We also provide the number of clusters returned by the density-based methods for the benchmark datasets in the revised Appendix (see Table 7). We can observe a similar improvement in recovering the ground truth across most datasets.
> > > >
> > > > **As it can be observed in Table 6, k-means provides better NMI results in 5 out 13 datasets which is a significant percentage.**
> > > >
> > > > Yes, we agree 5 out 13 datasets is a significant percentage. However, the benchmark datasets are mostly small-scale, low dimensional, with only a few clusters and not too many complex local features. See the meta information summarized in Table 4. As a result, the  advantage of local adapativity in the proposed methods is not that significant in these datasets. But still we can observe a very stable good performance across all datasets, and an uniform improvement comparing to classic density-based competitors.
> > > >
> > > > In more complex applications such as  clustering the face image datasets, the advantage of the proposed methods becomes more obvious.

---

> > > > > ### Comment · Reviewer_SSTf · 2021-11-29
> > > > > **Towards the end of the discussion phase, my recommendation does not change**
> > > > >
> > > > > I think that the authors have not adequately addressed my concerns.
> > > > > -It has been admitted that the method lacks theoretical justification and provides superior performance in the case of dense datasets with 'many complex local features'. However, only two datasets (related to faces) of this type have been considered in the experiments. Moreover, how one should know about the existence of 'many complex local features' in order to prefer the method over alternatives?
> > > > > -In order to achieve improvement over typical density-based methods, additional hyperparameters are used which are tuned in a supervised way taking into account the ground truth. Tuning a hyperparameter without ground truth knowledge is not an easy task, therefore hyperparameters should be kept to a minimum number.
> > > > > - I don't see any reason why \rho_{FKD} should perform better than \rho_{KD} as happens in several experiments. The performance difference is significant in some cases. Moreover, there is no consistent trend: in some cases  \rho_{FKD} performs better, while in other cases \rho_{KD} performs better. I don't like this inconcistency.

---

### Official Review · Reviewer_BUbF · 2021-10-29

**Correctness:** 2
**Technical Novelty And Significance:** 2
**Empirical Novelty And Significance:** 2
**Recommendation:** 5
**Confidence:** 4

**Details Of Ethics Concerns:**

The application of face clustering raises ethical issues with regard to its usage in surveillance.

**Main Review:**

In some regards, the motivation of the proposed density function is clear. The performance of density-based clustering methods depends to a large degree on the employed density measure and the traditional measures have obvious flaws. The proposed idea is sound and the experiments indicate that the proposed measures are able to improve the performance of clustering methods.

I am missing in this paper though the connection between the employed clustering methods (DBSCAN and DPC) and the density measure. Likewise, the connection of the proposed method to spectral clustering using the random walk Laplacian is not discussed at all. Shouldn't spectral clustering with the random walk Laplacian return a clustering which maximizes the stationary distribution function within each cluster as well[1]? The relation between spectral clustering and DBSCAN has been established [2,3]. Likewise, spectral clustering itself has an interpretation as a density-based clustering method[4]. As a result, since there is already some literature about kernel diffusion maps and spectral clustering (as cited by the authors), the relationship between these clustering methods and what is particularly novel in this work should be made clear.

For example, Theorem 1 states that clusters which are disconnected in the graph indicated by the epsilon neighborhood kernel have the same density in each cluster. From best practice in spectral clustering we know that disconnected clusters are very sensitive to noise and other perturbations. So, the case which is assumed in Thm 1 to motivate the proposed density measure, should actually be avoided. In this regard, I would ask for a more complete picture of density-based clusterings (including spectral clustering), their theoretical background, and how the proposed density measure can improve the clustering performance (also in dependence of the employed clustering method). At the same time, there exist methods to learn the kernel function such that the data is well clusterable. How is the proposed density measure positioned with respect to these methods?

The experimental evaluation should be made stronger. The authors state that the required kernel parameters are tuned, which indicates for me that the experimental evaluation makes use of the class information which is not available in real-life clustering applications. I think that this approach could be interesting for experiments such as the ones in Tbl1, where the potential of the density function is evaluated. However, there should be more experiments which simulate the actual clustering process. I would propose to use synthetic data to illustrate differences in the clustering behavior, where the clusters are generated according to given density measures. Such an analysis could also provide insight into the cases where the proposed measures do not work well. Another interesting analysis would be an experiment on synthetic data where the noise is increased. Then, also other clustering procedures should be compared, e.g. spectral clustering and newer nonconvex clustering methods.  How the parameters are defined in the face clustering application is not explicitly described, so I assume that again the parameters have been tuned, which should be avoided.

[1] Von Luxburg, Ulrike. "A tutorial on spectral clustering." Statistics and computing 17.4 (2007): 395-416.

[2] Y. Chen, "DBSCAN Is Semi-Spectral Clustering," 2020 6th International Conference on Big Data and Information Analytics (BigDIA), 2020, pp. 257-264, doi: 10.1109/BigDIA51454.2020.00048.

[3] Schubert, Erich, Sibylle Hess, and Katharina Morik. "The relationship of DBSCAN to matrix factorization and spectral clustering." LWDA. 2018.

[4] Hess, Sibylle, et al. "The SpectACl of nonconvex clustering: a spectral approach to density-based clustering." Proceedings of the AAAI Conference on Artificial Intelligence. Vol. 33. No. 01. 2019.

**Summary Of The Paper:**

The authors propose a new density function for density clustering models such as DBSCAN and Density Peaks Clustering (DPC). Given a kernel $K$ and the corresponding normalized random walk transition matrix $P = diag(K\mathbf{1})^{-1}K$, the authors propose to use the density function which corresponds to the stationary distribution attained in the limit. Since this function is expensive to compute, the authors propose a surrogate density function which is easier to compute.

The main experiments compare various density functions for DPC on UCI classification datasets and the DPC on density functions to face detection methods on two face identification datasets.

**Summary Of The Review:**

The authors propose a potentially interesting density measure for density-based clustering methods. Yet, the proposed measure is not sufficiently discussed in the scope of similar works for spectral clustering and the experimental evaluation is using class-label information for hyperparameter tuning, which is not eligible for clustering analyses. Hence, I reject the paper as is and suggest that the authors extend the experimental evaluation and provide a comparison to the related spectral clustering approaches for the next submission.

---

> ### Author Response · Authors · 2021-11-16
> **Response to Reviewer BUbF (Regarding the connection to spectral clustering)**
>
> The procedure of a density-based clustering algorithm is to first generate/estimate a density function and calculate its value at each point, then find the data points that can be viewed as cluster centers or core sets, and finally assign remaining points to the clusters by connecting with the higher density points nearby. It seems there is a serious misunderstanding of this procedure and the associated terminologies that have been discussed throughout the paper.
> We hope you would re-evaluate your score as all your main concerns are regarding this point. We will address your questions below in detail.
>
> **Shouldn't spectral clustering with the random walk Laplacian return a clustering which maximizes the stationary distribution function within each cluster as well?**
>
> Spectral clustering with random walk Laplacian first computes the first $k$ eigenvectors, then clusters the eigenvectors by k-means into $k$ groups. Our approach derives the limit distribution of the graph diffusion as the density. It shares the similar spitir to the probability density function of the underlying data generating mechanism. What we did is to apply this density to the density-based clustering methods.
>
> **The relation between spectral clustering and DBSCAN has been established [2,3]**
>
> The established relationship between spectral clustering and DBSCAN in [2,3] is based on the reachability between data points. As exhibited in [3], the reachability of OPTICS is defined as
> $$\text{reachability}(x_{i}, x_{j}):=\max\{\text{dist}(x_{i},x_{j}), \text{minPts-dist}(x_{i})\}.$$
>
>
> Our approach aims to provide a better alternative to the underlying "density function" of DBSCAN, DPC, and any other density-based clustering methods. As we can see there is no obvious connection to the spectral clustering methods.
>
> **Likewise, spectral clustering itself has an interpretation as a density-based clustering method[4]**
>
> As exhibited in [4], spectral clustering can be viewed as maximizing the average cluster density. For a cluster/subgraph $S$, this cluster density can be defined as
> $$\delta(S, W)=\frac{\sum_{i,j\in S}W_{ij}}{|S|},$$
> where $W$ is the adjacency matrix. The cluster density can be viewed as the average intra-connection.
>
>
> However, the term "density" used in density-based clustering is the probability density function of the underlying data distribution. This is quite different from the cluster density defined in [4]. We believe this is the major point that caused the reviewer's misunderstanding.
>
> [1] Von Luxburg, Ulrike. "A tutorial on spectral clustering." Statistics and computing 17.4 (2007): 395-416.
>
> [2] Y. Chen, "DBSCAN Is Semi-Spectral Clustering," 2020 6th International Conference on Big Data and Information Analytics (BigDIA), 2020, pp. 257-264, doi: 10.1109/BigDIA51454.2020.00048.
>
> [3] Schubert, Erich, Sibylle Hess, and Katharina Morik. "The relationship of DBSCAN to matrix factorization and spectral clustering." LWDA. 2018.
>
> [4] Hess, Sibylle, et al. "The SpectACl of nonconvex clustering: a spectral approach to density-based clustering." Proceedings of the AAAI Conference on Artificial Intelligence. Vol. 33. No. 01. 2019.

---

> > ### Author Response · Authors · 2021-11-19
> > **We look forward to your further feedback!**
> >
> > Dear Reviewer BUbF,
> >
> > Thanks again for your constructive suggestions and comments. As the deadline of discussion is approaching, we would like to know if there is any additional clarification or explanations that you may need, and we will be happy to provide them accordingly.
> >
> > In our previous response, we have studied your comments and addressed them carefully. We summarized the main points below:
> > 1. We provided  explanation on the differences between the proposed method and spectral clusting.
> > 2. We added a numerical study on comparison to spectral clustering, based on the bench datasets.
> > 3. We provided explanation on the hyperparameter issues. You may also want to have a look at our discussoins with Reviewer Nbci regarding similar issues.
> >
> > Please do not hesitate to let us know if there is additional response we can offer.
> >
> > Thank you for your time and effort spent in reviewing our paper!

---

> > > ### Comment · Reviewer_BUbF · 2021-12-01
> > > **Final Feedback**
> > >
> > > I find the idea of the paper quite cool, and as far as I see it's also correct. However, the experiments are just weak because of the hyperparameter tuning and there is no synthetic data used, to demonstrate emperically what the method is claimed to provide (generating the data according to the motivating case). In particular, I would like to respond to the following comment in the rebuttal:
> > >
> > > > A less effective method might be given the wrong credit due to a better parameter tuning.
> > >
> > > This is why at least some kind of heuristic should be provided to set the parameters. Follow-up researchers can then use these heuristics to compare to the new method. Ideally, you try to find good parameters for competitors but the heuristic is used to set your parameters. If you can then show an improvement over your competitors (not necessarily on all data but on the data which fits the motivation for your work at least), then this makes for a strong argument.
> > >
> > > There have been some improvements, such as the addition of SC to the experiments and also the parameter setting is clarified with regard to the face data. The connection to spectral clustering and the added novelty is still largely unclear. I take from the authors' response that their method is at least quite different to the cited reference (Nadir et al). I did not get a real response about my concerns of the Theorem assumptions. All in all, I would really like to see this paper in a polished and more round version.
> > >
> > > I'll increase my score to weak reject since I don't think anymore that there are no severe no gos in the paper after the rebuttal (the face data hyperparameters are at least not tuned).

---

> > ### Comment · Reviewer_BUbF · 2021-11-23
> > **Spectral Clustering Connection**
> >
> > Dear authors,
> > thank you for addressing my concerns, but it seems like we are talking past each other. You mention in your paper a reference (Nadler et al 2005) where diffusion maps are discussed with relation to spectral clustering. I was saying that you should explain your approach also with respect to possible applications in spectral clustering, because spectral clustering is also a density-based clustering method with strong ties to DBSCAN. In particular, I wonder how your approach connects with the existing work you cite?
> > The relationship of Spectral clustering with DBSCAN is simply established over the objective to find points that are densely connected (this is discussed more in detail in the references I pointed out). You propose a new way to define the density, yielding, in the end, a kernel or similarity matrix.  Density in "density-based" clustering may address the probability distribution of the underlying data distribution, but in the end, you calculate with a matrix giving you distances or inversely similarities - the matrix $W$.
> >
> > Do you agree with this description and if so, could you please elaborate on the connections and differences to (Nadlre et al. 2005)?

---

> > > ### Author Response · Authors · 2021-11-29
> > > **Difference to Nadlre et al. 2005**
> > >
> > >
> > > Thank you for your further comments. We would like to take this opportunity to clarify the fundamental difference between the proposed method and spectral clustering through diffusion maps, such as Nadlre et al.(2005).
> > >
> > >
> > > Diffusion maps leverages the relationship between heat diffusion and a random walk (Markov Chain) on a weighted graph whose nodes are sampled from the data. It maps coordinates between data and diffusion space, aims to reorganize data according to a new metric (diffusion distance). For example, if we look into diffusion maps, time $t$ is a crucial hyperparameter (which is equivalent to the bandwidth in KDE) in the diffusion density $\rho(x,t)$. Small $t$ represents local random walk, where diffusion distances reflect local geometric structure. Large $t$ represents global random walk, where diffusion distances reflect large-scale connected components. After a carefully tuning of $t$, a spectral clustering can be performed through diffusion maps, based on the intuition that the diffusion distance describes a “perceptual similarity” of points, e.g., points within the same cluster have small diffusion distances while in different clusters have large diffusion distances. Meanwhile, in the proposed method, we are constructing a diffusion process with  dissimilar properties and using its limit density as a function for density-based clustering.  If we consider the limiting density $\rho(x)$ in diffusion maps, it trivially groups all data points into one large cluster. Moreover, the local adaptive kernels that have been proposed in this paper do even satisfy the definition of kernels in classic settings, which is quite different from the heat kernel used in diffusion maps.

---

> ### Author Response · Authors · 2021-11-16
> **Response to Reviewer BUbF (Regarding Theorem 1 and Numerical Experiments)**
>
> ## Theorem 1
> The naïve density is defined as:
> $$\rho_{\text{naive}}(x) = \dfrac{1}{n \varepsilon^d}\sum_{y\in D} \dfrac{\textbf{1}_{B(x, \varepsilon) }(y)}{V_d},$$
> which is proportional to the number of data points covered by the $\varepsilon$-neighborhood around $x$. Since $\varepsilon$ is identical for every data point, the naïve density in large and compact clusters will become much larger than that in small and spread clusters, which makes it hard to discover small cluster centers. See our Figure 1 as a simple illustration. Theorem 1 shows that no matter how the points are distributed, the proposed densities of different points are now always comparable.
>
> ## Experimental evaluation
> **About the benchmark datasets**
>
> We agree that it is good to also evaluate the numerical performance on synthetic data. However, only using real-world data with complex features is quite standard in unsupervised learning papers.
>
> The benchmark datasets were also used in the paper of local contrast density [5] and many other relevant works. We consider it is better to apply our method to the same collection of the datasets.
>
> **About the tuning parameters**
>
> Just like k-means and spectral clustering that requires the  number of clusters as input, there are tuning parameters in density-based clustering methods such as DBSCAN and DPC. Tuning these hyperparameters properly is a necessary step to evaluate the clustering performance of all these methods.
>
> In $\rho_{naïve}$, $\rho_{LC}$, $\rho_{KD}$ and $\rho_{FKD}$, all hyperparameters were tuned in the same manner, just as in the  paper of local contrast density [5], which is one of our main competitors. Furthermore, in face image clustering, deep learning based methods such as CDP, L-GCN, LTC and GCN(V+E) are all tuned based on supervised measures. This is a common practice in existing works and we carried out experiments in the same way for a fair comparison.
>
>
> Additionally, as our methods do not rely on estimating the density via KDE, where bandwidth selection is very sensitive and  crucial. Sensitivity analysis is presented in Figure 3, where We can see the hyperparameters used in $\rho_{KD}$ and $\rho_{FKD}$ are less sensitive compared to other methods. And in this way, the selection of hyperparameters in our methods is much easier than other density-based methods.
>
>
>
>
>
> [5] Bo Chen, Kai Ming Ting, Takashi Washio, and Ye Zhu. Local contrast as an effective means to robust clustering against varying densities.Machine Learning, 107(8):1621–1645, 2018

---

> > ### Comment · Reviewer_BUbF · 2021-11-23
> > **Experimental Evaluation**
> >
> > Dear authors,
> > I have seen that you already discussed the tuning parameters issue with Rev Nbci and that for example the parameters in the face clustering application are not tuned. However, also the other experiments should not involve hyperparameter tuning. There is a difference between choosing an integer in k-means - for which also a lot of literature exists to set this parameter automatically- and tuning a combination of continuous parameters. What are then the experiments worth for practical applications? If [5] tunes the parameters in unsupervised applications then that is just bad practice and should certainly not be replicated.

---

> > > ### Author Response · Authors · 2021-11-29
> > > **Response to optimal parameter tuning**
> > >
> > > We agree with the reviewer that hyperparameter tuning in k-means is much easier compared to many state-of-the-art clustering methods. Actually, this was reflected in our numerical results in the revised paper (e.g., see Table 2 and 3), where $k$-means always pick the correct number of clusters. We also would like to mention that it is a common practice in the unsupervised learning community (not just [5]) that optimal tuning was carried out for complex data experiments (see all the recent competitor methods we cited in this paper). Although we agree this is not the same case as in real applications where no label information is available, thus is problematic to some extend. It is a good way to see how well different unsupervised learning methods can achieve in those datasets. Otherwise, it will be very difficult to fairly compare the performance of methods. A less effective method might be given the wrong credit due to a better parameter tuning. Also, people fore sure would argue for different results on the same method applied to the same dataset, but with different implementations of parameter tuning schemes. Therefore, it leaves us with no choice if we want to compare the proposed method with state-of-the-art works on face image clustering.

---

### Official Review · Reviewer_Nbci · 2021-10-31

**Correctness:** 2
**Technical Novelty And Significance:** 3
**Empirical Novelty And Significance:** 2
**Recommendation:** 5
**Confidence:** 4

**Main Review:**

The proposed method is interesting, and the presentation in the paper is reasonably clear, with some exceptions which I will note below. The empirical results certainly exhibit that potential for strong performance may be there, but I have reservations...

My primary concerns relate to the experimental set up and some uncertainties relating to the presentation, and also the clarity in relation to some of the technical points.

Experiments: -It isn't clear to me if the locally adaptive kernels were only used within the proposed approach, or also within DBSCAN and DPC. If only in the proposed approach, then it is not clear if the improvements in performance are actually a result of the new method, or simply down to the choice of kernel. I appreciate that the asymmetric kernel may not be usable in these frameworks, but simple adjustments can be used to symmetrise if necessary.
 -Without a data-driven means for selecting the tuning parameters for the model, it is not clear whether the results presented are realisable in practice. While "best case" plus "sensitivity study" do provide some evidence of practical relevance, it still isn't entirely clear how well we can expect the method to perform without access to an oracle tuner.
 -In relation to the face clustering problem, were the purpose-driven methods also given access to oracle tuning, or were they tuned using the data? If the latter, then these performance comparisons are not indicative of the actual comparative performance.
- The method is arguably more similar to normalised spectral clustering than to density clustering. Including results from spectral clustering seems an important change to make. In particular, if you use the same locally adaptive kernels within spectral clustering, does your method outperform?

Clarity: - The object D is unclear. At first this is the data set, and then later becomes the entire input space, as far as I can tell. This is especially important when included in technical statements like Assumption 1 and Theorem 2, where the interpretation might differ considerably.
- In the theoretical discussion, the objects \rho_{KD} and \rho_{FKD} are treated as constant, yet these are random variables. Is the convergence described in Theorem 2 sure convergence? As far as I can tell from the proof, it is, but the proof is not very clearly presented.
- The way Theorem 2 is stated, it seems to describe pointwise convergence rather than uniform convergence, yet uniform convergence is used in the text. Can you clarify?
- In Assumption 1 the value of c needs to be independent of n, presumably, otherwise a sequence tending to one as n tends to infinity might present a problem.
- You state that \rho_{FKD} can be determined in linear time. However, doesn't construction of P require at least log-linear time (or even worse?)

If these concerns can be suitably addressed, I would be willing to adjust my score upwards.

* Note that my score on "correctness" below is motivated by the fact that, without further evidence, the claims about practical performance are, in my opinion, not adequately supported. Otherwise the paper appears to be correct.

Some minor comments/questions/corrections follow:
- Why is a standard kernel density estimate referred to as ``naive''? Perhaps ``standard'' is more appropriate?
- The sentence "Such density suffers from capturing local features in complex datasets." in the abstract seems to be incorrect. Should this not read "Such a density suffers from an inability to capture local features in complex datasets.", or similar?
- You mention the issue of boundary bias in kernel density estimation. Does this have any relevance in density clustering?
- In Figure 1 I would argue that it isn't clear the diffusion density shows three clusters "clearly". Arguably the most sparse of the clusters actually manifests as multiple modes and that overclustering might result.
- The operator T_F seems to take x as an argument, so why is this suppressed in the definition?

Finally, although the grammar does not lead to difficulty in comprehension, I would recommend that the authors have their submission proof-read by a native English speaker as there are numerous minor grammatical imprecisions.

**Summary Of The Paper:**

The paper introduces a new approach to performing clustering, based on constructing a Markov process on the input space, in which transitions are determined according to normalised kernel similarity scores, as in Spectral Clustering (SC). Unlike SC, however, the proposal utilises the stationary distribution of the process, or an approximation thereof, evaluated on the data set, as a density function to be used within existing density based clustering algorithms, such as DBSCAN and Density Peaks Clustering (DPC). Experimental results are given to illustrate the methods potential improvement of standard kernel density estimates, and also its potential improvement over purpose-built models for face clustering.

**Summary Of The Review:**

An interesting idea with a fairly thorough discussion. However, some issues about clarity in relation to the technical parts of the paper.

Experimental results illustrate some promise, but it is very unclear if the reported performance is practicable as they are based on oracle tuning of hyper-parameters.

---

> ### Author Response · Authors · 2021-11-16
> **Response to Reviewer Nbci (Major Comments)**
>
> Thanks for your feedback. We hope the following point-by-point response addresses your concerns.
>
> **It isn't clear to me if the locally adaptive kernels were only used within the proposed approach, or also within DBSCAN and DPC.**
>
> Similar to linear KDE and local contrasting, the proposed kernel-diffusion approach focuses on developing a new density function, which shares the same spirit as the probability density function of the underlying data generating mechanism. The density can be applied to any density-based clustering methods, such as DBSCAN and DPC. The locally adaptive kernels were used to generate our kernel diffusion density. It can be seen as a pre-step of DBSCAN and DPC.
>
> **Without a data-driven means for selecting the tuning parameters for the model, it is not clear whether the results presented are realizable in practice.**
>
> In all the experiments, each hyperparameter in $\rho_{naïve}$, $\rho_{LC}$, $\rho_{KD}$ and $\rho_{FKD}$ was tuned in the same manner, just as in the paper of local contrast density [1], which is one of our main competitors. Furthermore, in face image clustering, deep learning-based methods such as CDP, L-GCN, LTC, and GCN(V+E) all include tuning parameters based on supervised measures. This is common in almost all the existing works and we carry out the same setting for fair numerical comparison. Additionally, as our methods do not rely on estimating the density via KDE, where bandwidth selection is very sensitive and crucial, choosing hyperparameters in the proposed methods is much easier compared to classic density-based clustering. We agree that in real-world applications, we have to select the tuning parameters in a data-driven manner.
>
>
>
>
> **The method is arguably more similar to normalized spectral clustering than to density clustering. Including results from spectral clustering seems an important change to make. In particular, if you use the same locally adaptive kernels within spectral clustering, does your method outperform?**
>
> We respectfully disagree with the reviewer regarding this point. Although both the proposed approach and spectral clustering require the construction of random walk Laplacian, their motivations and associated methodologies are completely different.
>
> Spectral clustering with random walk Laplacian first computes the first $k$ eigenvectors, then clusters the eigenvectors by k-means into $k$ groups. Our approach derives the limit distribution of the graph diffusion as the density. It serves a similar role to the probability density function of the underlying data generating mechanism. What we did is to apply it to the density-based clustering methods.
>
> We do not see a strong connection between these two methods. From our perspective, there is no reason to use the proposed kernels in Equations (4) and (5) in spectral clustering, and we can not see why improvement is expected, as the local adaptivity of the proposed kernels is coming from applying them to the diffusion process.
>
>
>
> **The object $D$ is unclear. At first, this is the data set, and then later becomes the entire input space, as far as I can tell.**
>
> The object $D$ always denotes the dataset and never becomes the entire input space. Note that in the paper, if an integral is on $D$, it is always with respect to an empirical (discrete) measure $F_n(\cdot)$.
>
> **In the theoretical discussion, the objects $\rho_{KD}$ and $\rho_{FKD}$ are treated as constant, yet these are random variables. Is the convergence described in Theorem 2 sure convergence? As far as I can tell from the proof, it is, but the proof is not very clearly presented.**
>
> **The way Theorem 2 is stated, seems to describe pointwise convergence rather than uniform convergence, yet uniform convergence is used in the text. Can you clarify?**
>
> Thank you for your suggestions. Yes, in Theorem 2 it is almost sure convergence. We have revised the theorem and its proof to make this clear. We also replaced the word "uniformly approximation" in the revised version to avoid any confusion.
>
>
>
> **In Assumption 1 the value of $c$ needs to be independent of $n$**
>
> Yes, the value of $c$ is independent of $n$. We emphasize this in the revised version.
>
> **You state that $\rho_{FKD}$ can be determined in linear time. However, doesn't construction of $P$ require at least log-linear time (or even worse?)**
>
> The construction of $P$ is needed in both $\rho_{\text{KD}}$ and $\rho_{\text{FKD}}$. This construction can be efficiently established [2], i.e. only takes a few seconds for more than a million data points. The more time-consuming step in practice is the computation of $\rho$. As a result, we always refer to the running time after the construction of $P$.
>
> [1] *Local contrast as an effective means to robust clustering against varying densities.Machine Learning*.  Chen, B., etc.  Machine Learning, 2018
> [2] Johnson, Jeff, Matthijs Douze, and Hervé Jégou. *"Billion-scale similarity search with gpus."* IEEE Transactions on Big Data (2019).

---

> > ### Comment · Reviewer_Nbci · 2021-11-17
> > **Resonse to response**
> >
> > 1. It seems the authors may have misunderstood my first point. I fully understand what has been done with THEIR method, I was curious about whether or not a standard KDE with the locally adaptive kernels and DBSCAN or DPC was done. If not, then it isn't clear if any improvements in performance are due to the different density estimator, or simply the different kernel being used.
> > 2. "choosing hyperparameters in the proposed methods is much easier": This needs to be justified if true.
> > 3. Spectral clustering vs proposed: I fully understand how spectral clustering works, and I believe I understand how the proposed method works. However, similarities between two clustering models depend not on the algorithms, but on the formulation of the models. As far as I can tell, both spectral clustering and the proposed approach seek to find subsets of the sample in which a random walk will spend considerably periods before exiting. Whether one uses eigenvectors of the transition matrix and the other the stationary distribution is rather irrelevant if the way clusters are defined is so similar.
> > 4. "The object D always denotes the dataset": I was under the impression that the proposed method seeks to define a Markov process over the entire space and cluster based on the evaluation of the stationary distribution at the sample points themselves. Although practically these are essentially equivalent, is it in fact the case that the Markov process is defined only on the data set? If I am wrong, then should not all instances of "density" be replaced with "mass" when speaking about the limiting distribution used for clustering?
> > 5. In relation to Theorem 2, it may be that sure convergence rather than almost sure convergence is achieved.

---

> > > ### Author Response · Authors · 2021-11-19
> > > **Thank you for the quick feedback, we hope the following points provide a satisfying answer to your concerns**
> > >
> > > **1. I was curious about whether or not a standard KDE with the locally adaptive kernels and DBSCAN or DPC was done. If not, then it isn't clear if any improvements in performance are due to the different density estimator, or simply the different kernel being used.**
> > >
> > > Thank you for clarifying your point. Note that KDE is used in DBSCAN or DPC as an estimate of the undelying pdf. Even if we use other classic KDE kernels or adaptive ones such as variable-bandwidth KDE [1], put aside the problem of tuning varying multivariate bandwidths, what we can get is a hopefully better estimate of the underlying pdf. The existence of local features/structures still chanllenges density-based clustering methods (recall that in DBSCAN we need to decide cluster center, core points, and noises based on the height of density function). For example, small or less concentrated clusters are still difficult to discover, especially if they are next to a large and compact cluster, as shown in Figure 1.  We want to justify that the fail of DBSCAN and DPC is not just because they are not using other standard kernesl or adaptive kernel bandwidths. In complex datasets such as emore_2000k and MS1M, there are thousands of clusters with a large number of local features/structures. Using a different kernel or improve the parameter tuning scheme will not solve the problem, or at least will not contribute to a significant improvement like our methods.
> > >
> > > The proposed $\rho_{KD}$ is not an estimate of the underlying pdf, but a finer version which overcomes abovementioned difficulties and is especially suitable for density-based clustering tasks. It resembles local contrast function, but the latter is too artificial and is mainly focus on magnifying the signal of small clusters.
> > >
> > > **2. "choosing hyperparameters in the proposed methods is much easier": This needs to be justified if true.**
> > >
> > > Thank you for pointing this out. In our sensitivity analysis (Figure 3), the clustering results are very stable on a wide range of values in terms of hyperparameters. These parameters in standard KDE-related methods are known to be very sensitive. Since we are using a diffusion density but not KDE, we do not have the problem of choosing optimal bandwidth to balance the bias-variance trade-off in estimation. We agree a rigorous theoretical justification is helpful to understand this phenomenon better, but it is beyond the scope of this paper.
> > >
> > > As in our answers to your point 1, and to your last question in the other response, we want to emphasis that our promising numerical results are not due to a finer parameter-tuning. In the face image clustering, we simply fixed the hyper-parameters at reasonable values without worrying about tuning them.
> > >
> > > **3. Spectral clustering vs proposed diffusion method**
> > >
> > > We agree with the reviewer that spectral clustering and our approach are  relevant in the sense that both running a random walk on the graph for considerably periods before exiting. However, in the proposed approach we do not seek to find subsets during the diffusion process, but to reveal the local features/structures in the density gradualy as the process moving forward. We added the performance of spectral clustering  on the benchmark datasets as a comparison in the Appendix, Table 6. The proposed approach uniformly outforms spectral clustering.
> > >
> > >
> > >
> > > **4. "The object D always denotes the dataset": I was under the impression that the proposed method seeks to define a Markov process over the entire space and cluster based on the evaluation of the stationary distribution at the sample points themselves. Although practically these are essentially equivalent, is it in fact the case that the Markov process is defined only on the data set? If I am wrong, then should not all instances of "density" be replaced with "mass" when speaking about the limiting distribution used for clustering?**
> > >
> > > The Markov process is on the data set $D$. In relevant literature, $\rho(x,t)$ is known as "density" of the graph diffusion process at time $t$, see [2]. Moreover, although KDE is an estimator over the entire space,  as you said, clustering is only based on its value at $n$ data points. The proposed $\rho_{KD}$ serves the same purpose as KDE (restrict to $D$) in density-based clustering.  Therefore, to avoid confusions we keep the term "density function" in this paper.
> > >
> > > **5. In relation to Theorem 2, it may be that sure convergence rather than almost sure convergence is achieved.**
> > >
> > > Yes it is sure/pointwise convergence. As this concept is not used very often in convergence of random variables and has no actual payoff compared to almost sure convergence, we feel the latter is preferable.
> > >
> > > [1]  *Variable kernel density estimation.* Terrell, D. G. and Scott, D. W.   Annals of Statistics, 1992.
> > >
> > >
> > > [2]  *Diffusion maps, spectral clustering and eigenfunctions of fokker-planck operators.*  Nadle, B., et al., NIPS, 2005.

---

> > > > ### Comment · Reviewer_Nbci · 2021-11-29
> > > > **Spectral clustering results not present**
> > > >
> > > > The authors say that the results for spectral clustering are given in Table 6 in the appendix, but I cannot find a Table 6. Is this just me?

---

> > > > > ### Author Response · Authors · 2021-11-29
> > > > > **Table 6 in the Appendix**
> > > > >
> > > > > Dear Reviewer Nbci,
> > > > >
> > > > > As we checked Table 6 was included in the Apendix of the latest revised version. The results for spectral clustering were presented in the last column of Table 6.

---

> > > > > > ### Comment · Reviewer_Nbci · 2021-11-29
> > > > > > **Thanks**
> > > > > >
> > > > > > Thank you! I realise now my mistake, I missed the appendix in the main pdf and was looking at the appendix given as supplement.
> > > > > > Best

---

> ### Author Response · Authors · 2021-11-16
> **Response to Reviewer Nbci (Minor Comments)**
>
> **Why is a standard kernel density estimate referred to as naïve? Perhaps standard is more appropriate?**
>  The term "naïve" referred to using the simplest $\epsilon$-ball in the kernel density estimate. This is to distinguish from standard KDE using Gaussian or other common kernels.
>
>  **The sentence "Such density suffers from capturing local features in complex datasets." in the abstract seems to be incorrect. Should this not read "Such a density suffers from an inability to capture local features in complex datasets.", or similar?**
>
> Thanks for noting this. We have corrected it accordingly in this version.
>
> **You mention the issue of boundary bias in kernel density estimation. Does this have any relevance in density clustering?**
>
> Yes, boundary bias will lead to problems in density-based clustering, as the data near the finite endpoints of the support will be not well-clustered.
>
> **In Figure 1 I would argue that it isn't clear the diffusion density shows three clusters "clearly". Arguably the most sparse of the clusters actually manifests as multiple modes and that over-clustering might result.**
>
> Since we don't know the true number of clusters, some boundary points may be viewed as outliers for density-based clustering methods. In Figure 1, the density centers of the three main clusters are very clear. The small density modes are at the boundary of the distribution mixtures, which is reasonable and has little effect on the clustering results.
>
> **The operator T_F seems to take x as an argument, so why is this suppressed in the definition?**
>
> In the revised version, we add the argument of $x$ in the operator $T_F$.

---

> > ### Comment · Reviewer_Nbci · 2021-11-17
> > **Response to response**
> >
> > "Yes, boundary bias will lead to problems in density-based clustering, as the data near the finite endpoints of the support will be not well-clustered":
> > - It isn't clear to me exactly why this would be the case. Boundary bias in kde is associated, as far as I know, with discontinuities in the density which lead to over-smoothing at these discontinuities when using kde. This does not seem as though it would have substantial bearing on the clustering result. Furthermore, even if it does, I do not see any reason why the diffusion density overcomes this bias. Perhaps the authors can explain?
> >
> > "Since we don't know the true number of clusters, some boundary points may be viewed as outliers for density-based clustering methods. In Figure 1, the density centers of the three main clusters are very clear. The small density modes are at the boundary of the distribution mixtures, which is reasonable and has little effect on the clustering results.":
> > - I agree that the three "true" modes are visible, but, as the authors note, we do not know there are only three "true" clusters, and so the presence of additional modes may be problematic. Furthermore, the plots in Figure 1 for KDE are clearly dependent on the bandwidth. Is it not the case that there is a bandwidth for which the three "true" modes are picked up? Since the authors do not mind about additional modes, the same concession should be made for the KDE, where a smaller bandwidth which, presumably, will pick up on these modes, will also lead to additional modes from the sparse cluster just as the diffusion density has.

---

> > > ### Author Response · Authors · 2021-11-19
> > > **Thank you for the quick feedback, we hope the following points provide a satisfying answer to your concerns**
> > >
> > > **It isn't clear to me exactly why this would be the case. Boundary bias in kde is associated, as far as I know, with discontinuities in the density which lead to over-smoothing at these discontinuities when using kde. This does not seem as though it would have substantial bearing on the clustering result. Furthermore, even if it does, I do not see any reason why the diffusion density overcomes this bias. Perhaps the authors can explain?**
> > >
> > > The reviewer's understanding about boundary bias is correct. This over-smoothing will generally result in unreliable estimates at the discontinuities/boundaries, for example, peaks at the density boundaries, which might lead to spurious clusters.   We agree that a formal analysis would be very interesting. Actually the proposed diffusion density overcomes the boundary bias problem. If we know the domain of the data, say data is on a one-dimensional close interval $[0,1]$,  we just need to solve our density in Equation (4) with the initial conditions, and with the following Neumann boundary condition. This is similar to carrying out a boundary correction using the reflection method.
> > > $$
> > > \dfrac{\partial}{\partial x}\rho(x,t)\bigg\vert_{x=0}=\dfrac{\partial}{\partial x}\rho(x,t)\bigg\vert_{x=1}=0.
> > > $$
> > >  We refer to [1] for a similar argument.
> > >
> > >
> > >
> > > **I agree that the three "true" modes are visible, but, as the authors note, we do not know there are only three "true" clusters, and so the presence of additional modes may be problematic. Furthermore, the plots in Figure 1 for KDE are clearly dependent on the bandwidth. Is it not the case that there is a bandwidth for which the three "true" modes are picked up? Since the authors do not mind about additional modes, the same concession should be made for the KDE, where a smaller bandwidth which, presumably, will pick up on these modes, will also lead to additional modes from the sparse cluster just as the diffusion density has.**
> > >
> > >
> > > In Figure 1, it is the local features (a smaller and another less concentrated cluster next to a large and compact cluster) that makes classic density-based clustering methods problematic. Here, three modes are not significant for clusteirng purpose even in the true underlying density. The standard KDE is not suitable in this scenario not just because of the choice of bandwidth. To see this, we plot the naive density in 3D on a range of smaller bandwidths in Appendeix, Figure 5.
> > >
> > >
> > > Just like our answer to your point 1 in the reponse to reponse (major comments), standard KDE with variable (data adaptive) kernel bandwidths will recover the true density better, but is not as helpful as our diffusion density in terms of clustering.
> > >
> > >
> > >
> > >
> > > [1] *Kernel Density Estimation Visa Diffusion.* Botev, Z., et al. Annals of Statistics, 2010.

---

> ### Comment · Reviewer_Nbci · 2021-11-29
> **Towards the end of the discussion phase, only a marginal improvement in my assessment**
>
> I remain somewhat ambivalent about the paper: On the one hand, there are some interesting points to the method and there is some evidence of potential for practical relevance in the experimental results. The authors have also done a reasonable job of handling some of my initial concerns. However, some concerns/questions seem to have been side-stepped to some extent. For example, my question about boundary bias and why that is not an issue for the proposed method was handled by suggesting a modification of their implementation which they admit is very similar to a reflection method for kde. If the problem can be corrected in both kde and their method, why should I conclude that theirs actually doesn't suffer from the same issue in practice? Most importantly, my concerns about the manner in which the experimental results were conducted remain. The "oracle tuning + sensitivity study" approach is not completely devoid of merit, but it doesn't give a clear indication of actual practical performance. The authors insist that tuning their method IS indeed easier than the other density based methods, but if this is the case then I don't understand why their experiments weren't conducted under a practicable scenario (i.e., by using whatever method they use for tuning their own method in practice). Since other reviewers have raised concerns about the tuning of parameters, it seems an obvious change to make. If the authors feel very strongly about the results they have presented, then they could provide both sets.
>
> My assessment of the paper is slightly more positive than it was initially, however not enough to tip it over the line. I feel there is still too much uncertainty about the the actual practical performance of the method.

---

### Official Review · Reviewer_apj8 · 2021-11-02

**Correctness:** 2
**Technical Novelty And Significance:** 2
**Empirical Novelty And Significance:** 3
**Recommendation:** 3
**Confidence:** 3

**Main Review:**

The suggested kernels are easy to understand, and many experimental results are provided, including sensitivity analysis to hyperparameters and computational analysis.

However, the authors need to more clearly explain and motivate their approach.

First of all, Section 4.1 needs to be re-written much more carefully and accurately. For example:
-  "$p(x,y)$ can be viewed as a probability for a random walk on the dataset from point x to point y": I don't think random walk is the correct term here, as random walks can be paths of any lengths.
-  "view D as a graph": a graph is a set of nodes and edges. Right now, D is only a set of points/samples. What are the edges?
- If L is the normalized graph Laplacian, is it a $n \times n$ matrix? If so, what is $L \rho(x, t)$ in equation (4) with the output of $\rho$ being a scalar?
- Aren't equations (4) and (5) *first*-order differential equations?
- The approach should be better motivated, i.e. include the answer to what is the point of all this discussion about diffusion, in the context of clustering? The paragraph on the top of page 5 is probably the most important part of Section 4, but it's not easy to understand. Can the authors explain the second half of that paragraph in more detail, perhaps with a figure or a toy example?

Clarifications for other parts:
- There needs to be a more concrete description/explanation of the density-based clustering algorithms, i.e. DBSCAN and DPC, using the notations from this paper. The introduction, related work or preliminaries would be a good place.
- Section 4.3 is a little hard to follow. For example in page 6, "This shows that $\rho_{FKD}$ elevates the density of small clusters": what does this sentence mean, and how does it follow from Theorem 1? Are there any divide-by-zero problems that could happen with $\rho$?

For the experiments:
- Sections 5.1 and 5.2 are missing sufficient reproducibility information. How many times were the experiments conducted? Should there be error bars? What are the "suitable range in the parameter space"? These are information that should have been included in the appendix if page limit was the issue.
- In Sections 5.4, how doest the computational cost of $\rho_{FKD}$ compare to the other density functions such as $\rho_{naive}, \rho_{LC}$? Is there a computation-accuracy tradeoff?

Smaller problems that can be fixed with editing:
- The sentence "Letting $h \to 0$, the random walk..." doesn't make sense. It may have too many verbs?
- In page 5, "some intuitions that why..."
- In page 5, "process grouping of all the data..."
- $\hat{g}$ and $\hat{\rho}$ in the appendix are not defined.


**Summary Of The Paper:**

This paper proposes a different density function for popular density-based clustering algorithms, i.e. truncated symmetric and asymmetric Gaussian kernels for DBSCAN and DPC. It offers some diffusion-based argument for why that is a better density function for clustering than naive function, and provides a computationally more efficient surrogate function as well.
Performance of proposed method on many clustering experiments are provided.

**Summary Of The Review:**

This paper has promising experimental results, but is not ready for publication at this stage. It could become a stronger paper after major revision by motivating and explaning their approach more carefully.

---

> ### Author Response · Authors · 2021-11-16
> **Response to Reviewer apj8 (on Section 4)**
>
> We understand the reviewer's concerns and hope the following points provide a satisfying response.
>
> 1. **I don't think random walk is the correct term here, as random walks can be paths of any length.**
> 2. **...a graph is a set of nodes and edges. Right now, $D$ is only a set of points/samples. What are the edges?**
> 3. **If $L$ is the normalized graph Laplacian, is it a $n\times n$ matrix? If so, what is $L\rho(x,t)$ in equation (4) with the output of $\rho$ being a scalar?**
> 4. **Aren't equations (4) and (5) first-order differential equations?**
> 5. **The approach should be better motivated, i.e. include the answer to what is the point of all this discussion about diffusion, in the context of clustering? The paragraph on the top of page 5 is probably the most important part of Section 4, but it's not easy to understand. Can the authors explain the second half of that paragraph in more detail, perhaps with a figure or a toy example?**
>
> We respectfully disagree with the reviewer regarding the above concerns. The construction of random walk/diffusion on graphs is quite standard in all areas of spectral graph theory. The exact same set of terms such as "random walk", "second-order differential equation" have been widely used in many popular existing works on different applications, to name a few, see [1,2,3]. We include as many explanations as any of these works. It is slightly unexpected for us to receive strong criticism with a low mark in terms of correctness.  We believe the presentation in Section 4 is mathematically rigorous and concise.
>
> To answer the reviewer's questions in more detail:
>
> 1. A random walk on a graph is a process that begins at some vertex, and at each time step moves to another vertex, e.g., the vertex that the random walk moves to is chosen with probability proportional to the weight of the corresponding edge among the neighbors of the present vertex. If we consider finer time steps in which smaller fractions of the probability leave the vertices. In the limit, this results in continuous random walks that can be modeled by the matrix exponential, as described in the paragraph before Equation (4). If we understand your question correctly, the "path of any length" in the classic random walk theory is just equivalent to "moves to any vertex on the graph" here.  This is an elegant analog of classic random walk theory in Euclidean space, where we can develop counterparts for concepts such as lazy walker, filtration, Markov property, and local limit theorems, etc. We refer to Section 10 in [4] for a basic-level introduction. This analog was adopted in numerous existing works. We do not see any problem with using the term "random walk" here.
>
> 2. Note that the transition probability matrix $P$ induces a weighted graph with vertices in $D$. Right after "view $D$ as a graph", we also mentioned that "(view) $L = I - P$ as the normalized graph Laplacian".
>
> 3. Yes, the graph Laplacian operator $L$ is a $n\times n$ matrix. As described in the paper, $\rho(x,t) \in D\times \mathbb{R}^+$ is the associated probability density, where the first argument takes different values in $\{x_1, x_2,\dots, x_n\}$. As a result, if $x=x_i$, $L\rho(x,t)$ is just the inner product of the $i$-th row in $L$ and the vector of $(\rho(x_1,t), \dots,\rho(x_n,t))^T$. This is a  standard notation in the literature, for example, see [3].
>
> 4. Equations (4) and (5) are second-order differential equations, as graph Laplacian corresponds to the second-order derivative. Note that we were not referring to the partial derivatives of $\rho(x,t)$ with respect to $t$ on the LHS of Equations (4) and (5). Please see [3, 5] for similar usage of terms.
>
> 5.  Above explanations might be helpful for the reviewer to understand better the discussion of the diffusion process on a graph. This is the core of our methodology. Using the limiting distribution of this diffusion, we developed a completely new framework of density with desired properties such as local adaptivity. This density can be applied to density-based clustering algorithms and improve their performance by a large margin. We have rephrased the paragraph on the top of page 5.
>
> We hope that this reply convinces you that our Section 4 is accurate and hope you would re-evaluate your score as these graph diffusion terminologies and notations seem to be your main technical concern.
>
> 1. *Geometric diffusions as a tool for harmonic analysis and structure definition of data: Diffusion maps.* Coifman, R. R., etc. PNAS, 2005.
> 2. *Graph Laplacians and their Convergence on Random Neighborhood Graphs.* Hein, M., etc. JMLR, 2007.
> 3. *Towards a theoretical foundation for Laplacian-based manifold methods.* Belkina, M. and Niyogib, P. Journal of Computer and System Sciences, 8(74), 1289-1308, 2008.
> 4. *Spectral and Algebraic Graph Theory.* Spielman, D. A. 2019.
> 5.  *Diffusion maps, spectral clustering and eigenfunctions of fokker-planck operators.*  Nadle, B., etc. NIPS, 2005.

---

> > ### Author Response · Authors · 2021-11-19
> > **We look forward to your further feedback!**
> >
> > Dear Reviewer apj8,
> >
> > Thanks again for your very helpful suggestions and comments. As the deadline of discussion is approaching, we would like to know if there is any additional clarification or explanation that you may need, and we will be happy to provide them accordingly.
> >
> > In our previous response, we have studied your comments and addressed them carefully. We summarized the main points below:
> > 1. We provided explanation on the graph diffusion process, and the associated terminologies and notations.
> > 2. We rephrased the intuition of the diffusion density in the revised paper.
> > 3. We made it clear that the numerical studies are real data analysis and provided the computational cost of the proposed approach.
> >
> > Please do not hesitate to let us know if there is additional response we can offer.
> >
> > Thank you for your time and effort spent in reviewing our paper!

---

> ### Author Response · Authors · 2021-11-16
> **Response to Review  apj8 (other comments)**
>
> **There needs to be a more concrete description/explanation of the density-based clustering algorithms, i.e. DBSCAN and DPC, using the notations from this paper.**
>
> Thank you for your suggestion. In the revised appendix, we added descriptions of the DBSCAN and DPC algorithms through pseudo code .
>
> **Section 4.3 is a little hard to follow. For example on page 6, "This shows that $\rho_{FKD}$ elevates the density of small clusters": what does this sentence mean, and how does it follow from Theorem 1? Are there any divide-by-zero problems that could happen with $\rho$?**
>
> In Theorem 1 we showed that "the average $\rho_{FKD}$ in each cluster are the same regardless of cluster sizes and other local features". This means $\rho_{FKD}$ has the effect of magnifying density values in small clusters, similar to the local contrasting. Thus when applying $\rho_{FKD}$ to density-based clustering, it becomes easier to identify small clusters.
>
>
> There is no divide-by-zero problem. The normalized graph Laplacian construction is well-defined.
>
>
>
>
>
>
>
> **Sections 5.1 and 5.2 are missing sufficient reproducibility information. How many times were the experiments conducted? Should there be error bars? What is the "suitable range in the parameter space"? These are information that should have been included in the appendix if page limit was the issue.**
>
> All the numerical studies in Sections 5.1 and 5.2 are clearly conducted on real-world datasets. They are not Monte Carlo simulations. As a result, we run each experiment once and there are no error bars to show. This is quite standard in most unsupervised learning works, where real-world data with complex features are much more challenging than synthesized data. We do not quite understand why the reviewer had concerns about reproducibility.
>
>
>
> We have now included the following information in the Appendix regarding the choice of parameters: The parameter $\varepsilon$ (radius of the ball, used in $\rho_{\text{naive}}$, $\rho_{\text{LC}}$, $\rho_{\text{KD}}^{\text{sym}}$ and $\rho_{\text{FKD}}^{\text{sym}}$) is tuned by searching within the range between 0.1 and 1 with am increment of 0.1, parameter $k$ (number of nearest neighbors, used in $\rho_{\text{LC}}$, $\rho_{\text{KD}}^{\text{asym}}$ and $\rho_{\text{FKD}}^{\text{asym}}$) is tuned by searching within the range between $0.1n$ and $0.5n$, with an increment of $0.1n$, where $n$ is the sample size.
>
>
> **In Sections 5.4, how does the computational cost of $\rho_{FKD}$ compare to the other density functions such as $\rho_{naive}$ and $\rho_{LC}$? Is there a computation-accuracy tradeoff?**
>
> $\rho_{FKD}$ has the same linear computational cost just as linear KDE methods such as $\rho_{naive}$ and $\rho_{LC}$. In small sample datasets, it is possible that $\rho_{FKD}$ and $\rho_{KD}$ have different performances, but there is no strict computation-accuracy tradeoff phenomenon.
>
>
> **Smaller problems that can be fixed with editing...**
>
> Thank you very much for pointing out these smaller problems. We have revised them accordingly.

---

### Decision · Program_Chairs · 2022-01-20

**Decision:**

Reject

**Comment:**

This paper proposes a kernel diffusion method to improve upon density-based clustering methods. The reviewers found the empirical results quite promising and there is consensus that there are some good ideas in this work. However, their criticisms are strikingly consistent that the technical details are lacking and some of the claims are not fully supported, and these criticisms were not found to be fully addressed in the author responses. I agree with the assessment that this is promising in a major revision toward a future submission but it is currently not complete, especially in the theoretical and technical details.